# Multi-area recordings and optogenetics in the awake, behaving marmoset

Patrick Jendritza [1,2] ✉, Frederike J. Klein[1] & Pascal Fries [1,2,3]

The common marmoset has emerged as a key model in neuroscience. Marmosets are small in size, show great potential for genetic modification and exhibit complex behaviors. Thus, it is necessary to develop technology that enables monitoring and manipulation of the underlying neural circuits. Here, we describe a novel approach to record and optogenetically manipulate neural activity in awake, behaving marmosets. Our design utilizes a light-weight, 3D printed titanium chamber that can house several high-density silicon probes for semi-chronic recordings, while enabling simultaneous optogenetic stimulation. We demonstrate the application of our method in male marmosets by recording multi- and single-unit data from areas V1 and V6 with 192 channels simultaneously, and show that optogenetic activation of excitatory neurons in area V6 can influence behavior in a detection task. This method may enable future studies to investigate the neural basis of perception and behavior in the marmoset.

The common marmoset (*Callithrix jacchus*) is becoming an important animal model in neuroscience[1-4]. Due to their small size, genetic tractability[5-7] and rich behavioral repertoire[8-10], marmosets hold great potential for improving our understanding of the neural circuits underlying complex behaviors and perception. It is therefore pivotal to develop techniques that enable monitoring and manipulation of these circuits in awake, behaving animals.

Extracellular single-unit recordings remain an essential method in systems neuroscience due to their unparalleled temporal resolution and ability to record from almost any location in the brain[11]. While earlier studies mostly utilized tungsten microelectrodes[12-14], the use of silicon-based microelectrode arrays has recently been established in awake marmosets[15-18]. These contributions have paved the way for better access to the neural circuits of the marmoset brain.

The characterization of response properties of neurons from the visual cortex of the marmoset is almost entirely based on experiments performed under anesthesia (for comprehensive reviews, see refs. [2,8]). In contrast, data from visual areas in awake marmosets is still very scarce[15,16,19]. Even more strikingly, there is only one study of single-unit recordings in awake marmoset primary visual cortex (V1)[19], in stark contrast to the wealth of studies on this area in other species. Hence,

the relative lack of published work in awake animals emphasizes the need to develop suitable recording approaches.

Importantly, beyond the correlative evidence that can be obtained from recordings, manipulation of neural activity can be used to gain insight into the causal link between neural circuits and behavior[20]. Optogenetics is a powerful tool for such questions, because it offers the necessary spatiotemporal and genetic precision[21]. The principal feasibility of optogenetic stimulation techniques in marmosets has already been demonstrated[22-24]. However, the integration of neural recordings, optogenetics and behavioral manipulation is still lacking.

Here, we demonstrate a method to perform neural recordings with high-density silicon probes from two visual areas simultaneously and show for the first time that optogenetic stimulation of area V6 can influence the animal's behavior in a detection task.

## Results

### Implant design and recording approach

Our goal was to design a small and lightweight implant that utilizes modern high-density silicon probes while providing access to optogenetic stimulation techniques in awake-behaving marmosets.

[1]Ernst Strüngmann Institute (ESI) for Neuroscience in Cooperation with Max Planck Society, Frankfurt, Germany. [2]International Max Planck Research School for Neural Circuits, Frankfurt, Germany. [3]Donders Institute for Brain, Cognition and Behaviour, Radboud University Nijmegen, Nijmegen, The Netherlands. ✉e-mail: patrick.jendritza@esi-frankfurt.de

The complete implant consists of multiple parts: Headpost, chamber, microdrives, stabilizers, silicon probes and printed circuit boards (PCBs) holding the connectors (Fig. 1a). The 3D-printed titanium chamber was designed to smoothly fit onto the surface of the marmoset skull (Fig. 1a, b). The chamber houses six PCBs with connectors, which relay the neural signals from two silicon probe arrays: A four-shank 4 × 32-channel silicon probe targeting visual area V6 and a two-shank 2 × 32 channel silicon probe targeting visual area V1, amounting to a total of 192 channels (Fig. 1a). Silicon probes are mounted to microdrives that allow for up to 5 mm vertical travel. Microdrives are supported by 3D-printed titanium stabilizers that provide additional rigidity after implantation. The stabilizers are designed to be positioned very close to the skull, such that they minimize the gap that needs to be filled with the cement during implantation (Supplementary Fig. 1). Thus, they make the implantation process easier and faster. The probes are implanted through a small (≈2 mm diameter) craniotomy (Fig. 1c) that is sealed with a transparent silicone gel[25]. Optogenetic stimulation can then be performed by pointing an optic fiber at the craniotomy such that the light penetrates through the silicone into the tissue (Fig. 1c).

To allow stabilization of the animal's head during recordings, a CNC-milled titanium headpost was implanted in front of the chamber (Fig. 1a, b). The inside of the chamber is protected by a 3D-printed nylon lid that can be secured by four small screws on the side of the implant (Fig. 1a, d). The height of the lid can be chosen depending on implant requirements. Fig 1b, shows a photograph of the chamber on a skull model with the flat version of the lid and the headpost in place.

We implanted chamber and headpost in five animals (Table 1). All animals tolerated the implant well, without necessity of post-implantation wound care. None of the lids required replacement, even after several months of use with almost daily opening and closing. Three of the five animals were subsequently implanted with electrodes in areas V1 and V6. Figure 1d shows a photograph of the final implant in Monkey A during the opening of the lid just prior to recording.

Size and weight minimization of an implant are important design factors when working with small animals. These factors are not only crucial in order to ensure the welfare of the animal, but also facilitate the study of natural behaviors[26,27].

The chamber was designed to span 28 mm in the anterior–posterior axis and 17 mm in the mediolateral axis of the skull

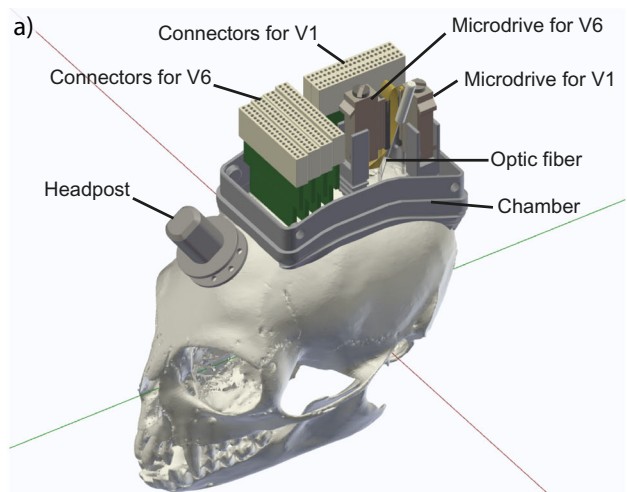

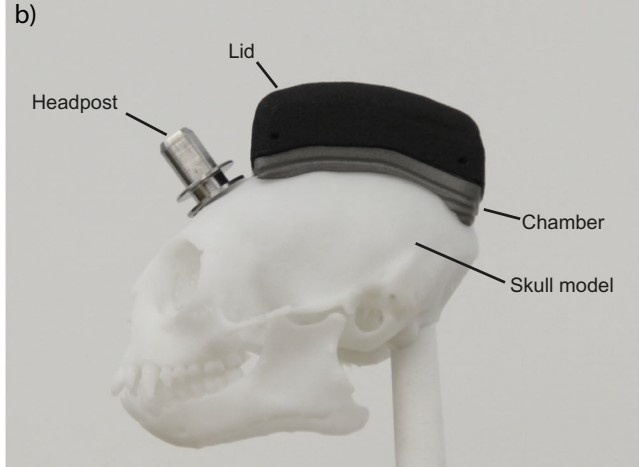

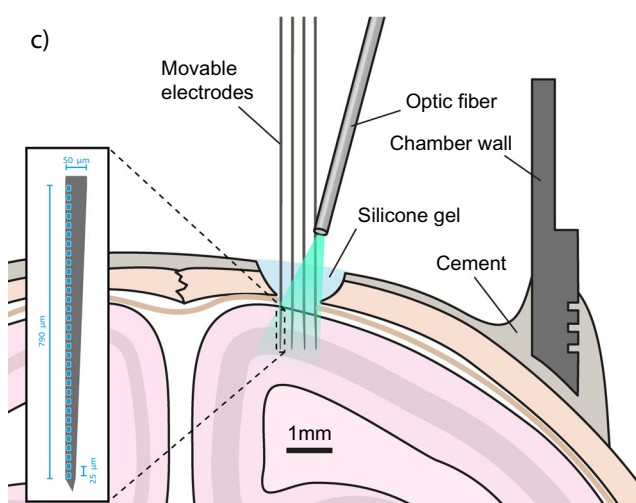

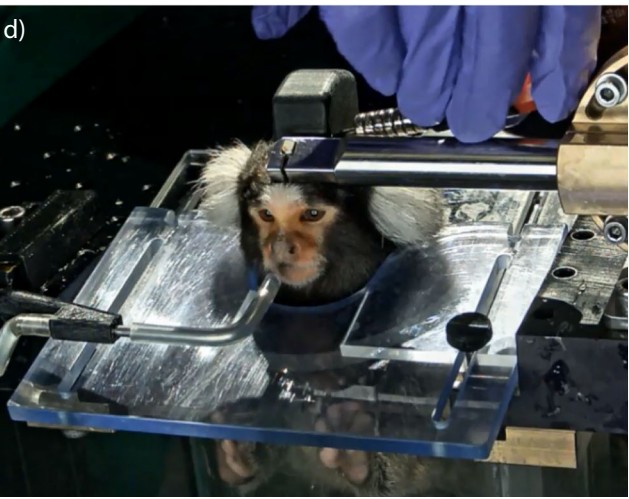

**Fig. 1 | Implant design and recording approach. a** 3D rendering of the complete 192-channel implant design. A four-shank silicon probe with 4 × 32 channels is attached to the microdrive targeting area V6. A two-shank silicon probe with 2 × 32 channels is targeting area V1. The four connectors at the anterior end of the chamber are wired to the probe in area V6. The two connectors at the posterior right side of the chamber are wired to the probe in V1. An optic fiber (200 μm diameter) is placed above the V6 craniotomy with an external micromanipulator that guarantees flexible and precise positioning (not shown for clarity). The headpost for stabilizing the animal during recording is placed in front of the chamber. **b** Side-view photograph of a skull model with headpost, chamber and flat lid as used after implantation of headpost and chamber. **c** Illustration of a coronal section of the target location in area V6. Craniotomy, electrodes and chamber are drawn to scale. Inset shows magnified view of electrode layout. **d** Photograph of Monkey A while head-fixed and facing the monitor, during opening of the tall lid used after electrode implantation. The photograph shows the animal with the final 192-channel implant.

**Table 1 | List of all animals, procedures and outcomes**

|  | Monkey A | Monkey U | Monkey D | Monkey E | Monkey P |
|---|---|---|---|---|---|
| Sex | Male | Male | Male | Male | Male |
| First surgery performed (headpost, chamber, ref. wire) | Yes | Yes | Yes | Yes | Yes |
| Body weight at first surgery | 385 g | 438 g | 455 g | 530 g | 428 g |
| Second surgery performed (electrodes, viral vector) | Yes | Yes | Yes | No | No |
| Body weight at second surgery | 371 g | 460 g | 445 g | n.a. | n.a. |
| Neural recordings in V1 | Yes | Yes | Yes | n.a. | n.a. |
| Neural recordings in V6 | Yes | Poor | Yes | n.a. | n.a. |
| Optogenetic stimulation in V6 | Yes | Poor | Yes | n.a. | n.a. |
| Duration (months) after first surgery* | 40 | 26 | 26 | 26 | 26 |
| Duration (months) after second surgery* | 35 | 19 | 19 | n.a. | n.a. |
| Data shown in figures | Fig. 1, Fig. 4, Fig. 5, Fig. 6, Fig. 7, Fig. 8 | – | Fig. 3, Fig. 7 | – | – |

*Relative to the time this manuscript was prepared (September 2021).

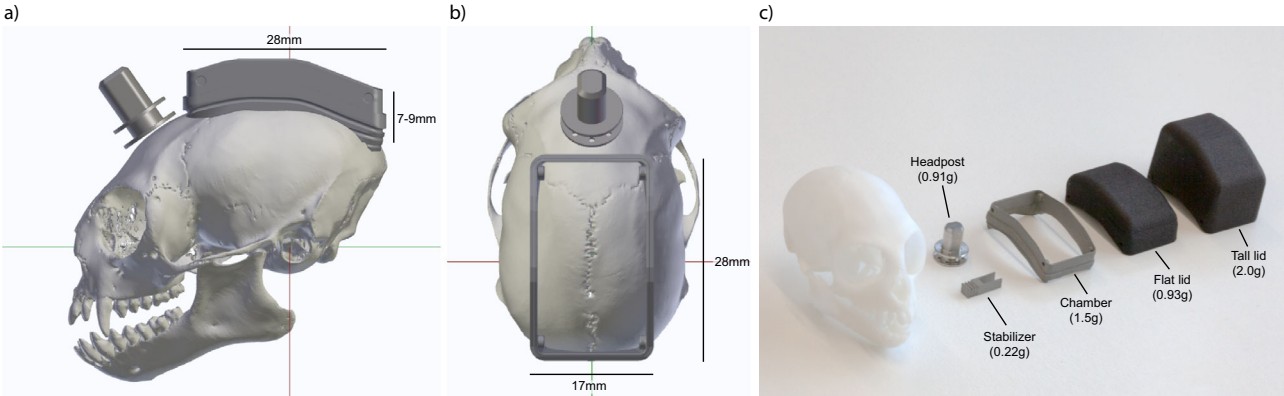

**Fig. 2 | Implant size and weight.** 3D rendering of side view (**a**) and top view (**b**) of a marmoset skull with headpost and chamber in target position, aligned in stereotaxic coordinates. Red line indicates interaural axis. Green line indicates anterior–posterior axis. **c** Photograph of the CNC-machined and 3D-printed parts of the implant next to a skull model. Weights are indicated in parentheses.

(Fig. 2a, b, outer chamber dimensions). We restricted the lateral extent of the chamber such that the implantation required only minimal detachment of the temporal muscle from the bone. The sides of the chamber extended laterally only 1–2 mm beyond the superior temporal lines of the skull. This design allows targeting a large number of dorsal brain areas for neural recording and stimulation (Supplementary Fig. 2).

The height of the final implant depends on the selection of electrodes and connectors inside the chamber. The chamber protrudes only 7–9 mm from the surface of the skull (Fig. 2a). When closed with the flat lid (e.g., without probes installed), it reaches a height of 12–14 mm from the skull. After implantation with silicon probes and connector PCBs as used here, the chamber is closed with a taller version of the lid, and the implant reaches a height of 20–22 mm from the skull.

Recent advances in 3D printing make it possible to accurately manufacture complex shapes from medical-grade titanium (Ti6Al4V). The mechanical strength of titanium allowed us to reduce the wall thickness of the chamber to 0.5–1 mm (Figs. 1c and 2b), which resulted in a weight of only 1.5 g (Fig. 2c). Headpost and stabilizers had a weight of 0.91 g and 0.22 g, respectively. Tall and flat lids were 3D-printed from polyamide (PA12 nylon) and weighed 0.93 g and 2.0 g, respectively. Thus, the total resulting weight of the implant was only ≈8 g, including headpost, chamber, silicon probes, microdrives, stabilizers, connectors and cement.

## Two-stage implantation procedure

In order to maximize chances of surgical success we adopted a two-stage implantation procedure and made use of customized 3D-printed implantation holders. First, headpost and chamber were implanted in the same initial surgery (Surgery 1). After appropriate recovery time, a second surgery was performed, in which a viral vector was injected and several silicon probes were implanted (Surgery 2).

## Surgery 1: Implantation of chamber and headpost

At the beginning of the first surgery, the animal was placed in a stereotaxic apparatus, and the skull was prepared for implantation (see "Methods"). Chamber and headpost could then be lowered onto the skull surface for alignment. Precise alignment of the chamber relative to the skull was crucial, because it ensured that the chamber could later be used as positional reference for the stereotaxic coordinate system. Both, chamber and headpost were held by a custom implantation holder that was attached to a micromanipulator (Fig. 3a). Prior to the surgery, markers on the sides of the holder were used for alignment to the interaural line (i.e., axis of the ear bars). This assured correct positioning of the chamber in the anterior–posterior axis. During the surgery, a downward-pointing wedge integrated into the holder was aligned to the central skull suture, to assure correct positioning in the mediolateral axis (Fig. 3a). After alignment, the position of the holder was locked, and the holder was temporarily removed to allow better access for the subsequent surgical steps.

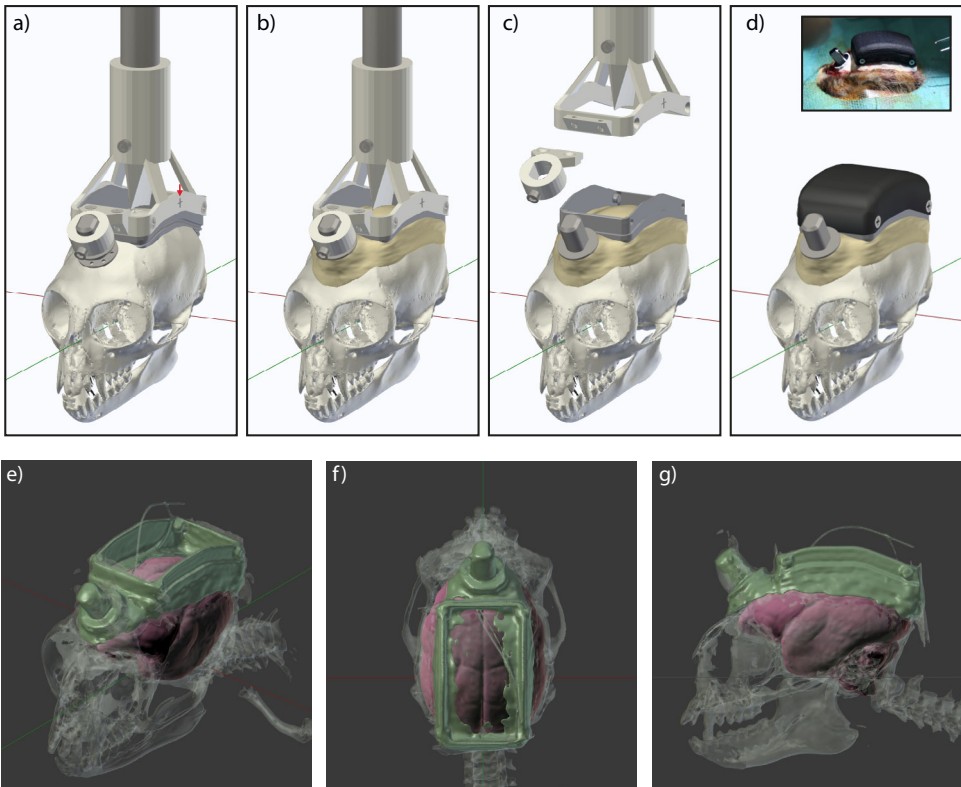

**Fig. 3 | Surgery 1: Implantation of chamber and headpost. a** Chamber and headpost were held by a custom implantation holder that was attached to a micromanipulator. Note the cross-shaped markers on the side of the holder, used for alignment to the interaural axis, prior to the surgery (red arrow). A wedge-shaped guide pointing downwards in the center of the holder was used for mediolateral alignment to the central skull suture. **b** Following skull preparation, the aligned chamber and headpost were cemented onto the skull. **c** Once the cement had hardened, chamber and headpost were released from the holder. **d** The chamber was closed with a 3D-printed nylon lid for protection. Inset shows photograph of the implant at the end of the first surgery. **e** Near-isometric projection, (**f**) top view and (**g**) side view of the 3D segmentation from a CT scan after the first surgery in Monkey D. Radio-opaque cement, metal parts and reference wires show the highest contrast and are colored in green. Bone is shown in semi-transparent gray. The fitted MRI-based template brain is shown in red. Note that the cement layer in the center of the chamber is very thin and therefore not visible everywhere in the segmented data.

---

The skull surface was then cleaned and coated with dental adhesive before a thin layer of cement was applied[28]. Two platinum wires were implanted epidurally anterior to the chamber, serving as backup reference wires. The actually used reference wires were implanted subdurally in the second surgery. Next, the implantation holder was returned to the previously determined anteroposterior and mediolateral position, and lowered until the chamber contacted the skull. The headpost and chamber were then cemented in place (Fig. 3b). After the cement had hardened, headpost and chamber were released from the holder (Fig. 3c). At the end of the surgery, the flat version of the lid was used to close the chamber (Fig. 3d). The animal was then allowed to recover for two weeks and subsequently underwent head-fixation training.

Variability in head morphology between animals can lead to inaccuracies during stereotaxic surgeries. Therefore, after the first surgery, we obtained anatomical data of the skull and implant via computed tomography (CT) scans (Fig. 3e–g). Appropriate thresholding of CT data allowed segmentation of the bone (shown in transparent gray), and of metal and radio-opaque cement (shown in green). After segmentation, the inside of the animal-specific skull model was used to fit an MRI-based template marmoset brain[29]. This approach can be justified under the assumption that the gap between bone and the brain is very small. Fits were performed manually by translating and scaling in all three spatial dimensions, and rotating in the pitch axis. The resulting fit of the template brain and its area delineations can then serve as individualized anatomical references for each animal. Thereby, we obtained the precise positions of our target areas in the same reference frame as the chamber visible in the CT. Note that this CT-based targeting refinement was only used in marmosets D and U.

## Surgery 2: Injection of the viral vector and implantation of silicon probes

To assure correct positioning in the second surgery, the implantation holder from the first surgery (Fig. 3a–c) was used to re-align the animal's head via the previously implanted chamber: After ensuring sufficient depth of anesthesia, the lid was removed, and the chamber attached to the animals' skull was re-inserted into the holder. This effectively re-aligned the skull of the animal to precise stereotaxic coordinates as defined by the holder and the chamber. Subsequently, a high-precision articulated arm was used to fix the animals' head position via the implanted headpost. After locking the articulated arm, the chamber holder was removed. Thus, the use of ear bars and eye bars could be avoided in the second surgery, thereby reducing potential discomfort for the animal.

Next, the inside of the chamber was disinfected with $H_2O_2$ and ethanol. A 3D-printed guide was temporarily placed on the chamber and used to mark the target positions for the craniotomies over areas V1 and V6 of the left hemisphere (Supplementary Fig. 3). Two platinum wires, serving as reference electrodes, were then implanted subdurally at the anterior end inside the chamber, through a small burr hole (≈2 mm diameter). Next, two small burr holes were made at the target locations for the electrodes over V1 and V6. A durotomy of ~1.5 mm was performed over area V6, and the viral vector was injected (Fig. 4a).

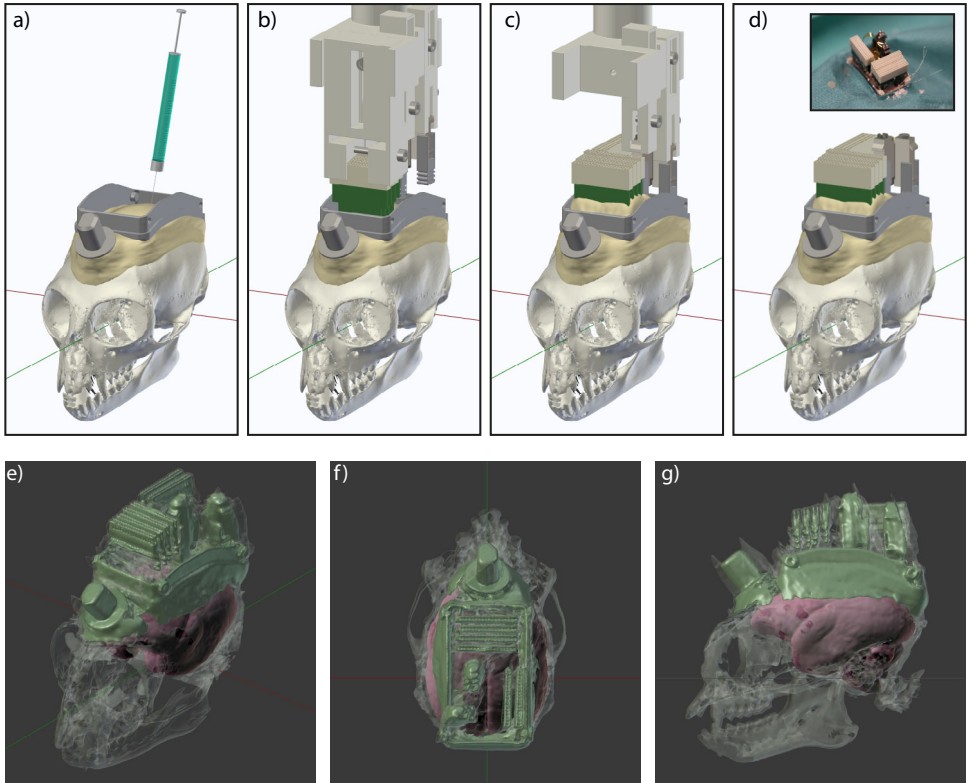

**Fig. 4 | Surgery 2: Injection of the viral vector and implantation of silicon probes. a** After stereotaxic alignment of the skull via the implantation holder and the chamber, a viral vector was injected into area V6. **b** A custom implantation holder, carrying connector PCBs, electrodes, and microdrives was lowered into the chamber. **c** First, the connector PCBs were cemented in place and the respective part of the holder was removed to ensure better access and visibility. Electrodes were then lowered sequentially into the two brain areas, and the respective microdrives were cemented into position. **d** After all parts were secured, the holder was completely removed. Inset shows a photograph at the end of the second surgery. **e** Near-isometric projection, (**f**) top view and (**g**) side view of the 3D segmentation form a CT scan after the second surgery in Monkey A. Radio-opaque cement and metal parts (including connectors and microdrives) show the highest contrast and are colored in green. The bone is shown in semi-transparent gray. The fitted MRI-based template brain is shown in red.

After a short waiting time for diffusion of the vector into the tissue, the needle was slowly retracted.

A 3D-printed implantation holder was then lowered into the chamber (Fig. 4b). The holder was prepared prior to the surgery to hold all necessary components for the implantation: two microdrives (with silicon probes and stabilizers attached) and six connector PCBs. The three main components (connector PCBs, V1 microdrive with probes and V6 microdrive with probes) were held by separate parts of the implantation holder, enabling independent movement in the *z* axis. This independence allowed sequential implantation of the components. To this end, the holder was initially prepared such that the connector PCBs were at the lowest position and were thus implanted first (Fig. 4b). Connector PCBs were positioned via the micromanipulator just above the cement layer on the skull, and were then cemented in place. After curing, the part of the implantation holder securing the connector PCBs was removed (Fig. 4c). This resulted in better visibility and allowed for independent movement of the microdrives holding the silicon probes (Fig. 4c). Next, the probe array for area V6 was implanted. In order to insert the silicon probe into the cortex at the optimal position relative to the durotomy and the local cortical vasculature, the anterio–posterior and mediolateral positions of the implantation holder were fine-tuned before probe insertion. After the probe was slowly inserted into the superficial part of the cortex (<500 μm), the microdrive with its attached stabilizer was cemented into the chamber. Subsequently, the part of the implantation holder that was securing the V6 microdrive was removed, too. The same procedure was performed for area V1, and the implantation holder was completely removed

(Fig. 4d). Both craniotomies were then sealed with silicone gel (Fig. 1c).

Animals recovered quickly after the second surgery and were brought into the recording setup within a few days. To visually inspect the position of the microdrives and PCBs, we obtained a CT scan from Monkey A after the second surgery (Fig. 4e–g). The high-contrast metal parts of the connectors and microdrives with stabilizers are visible in green color. Bone is shown in semi-transparent gray and the fitted MRI-based template brain in red.

## Simultaneous recording in areas V1 and V6

After slowly lowering the probes into the brain, clear spiking activity was visible across several recording sites in areas V1 and V6 (Fig. 5a).

In order to test visual responsiveness and spatial selectivity, we performed receptive field (RF) mapping with multi-unit activity (MUA). Flashing annulus and wedge stimuli were presented while the animal was maintaining its gaze on a central fixation point. A detailed account of the RF mapping procedure can be found in ref. [30]. As expected from the implantation target position, RFs in area V1 were located in the lower right visual field (Fig. 5b). Furthermore, RFs showed substantial overlap for all electrodes along a given probe shank (Fig. 5b, black outlines at the bottom).

Next, we presented static gratings to the animals. MUA following visual stimulation with gratings was visible across several recording sites and peaked shortly after stimulus onset (Fig. 5c, d). Sites were considered to be visually modulated if MUA between pre-stimulation baseline (−0.25–0 s) and poststimulus time (0 to 0.65 s) was significantly different ($P < 0.05$; two-sided Kolmogorov–Smirnov test for

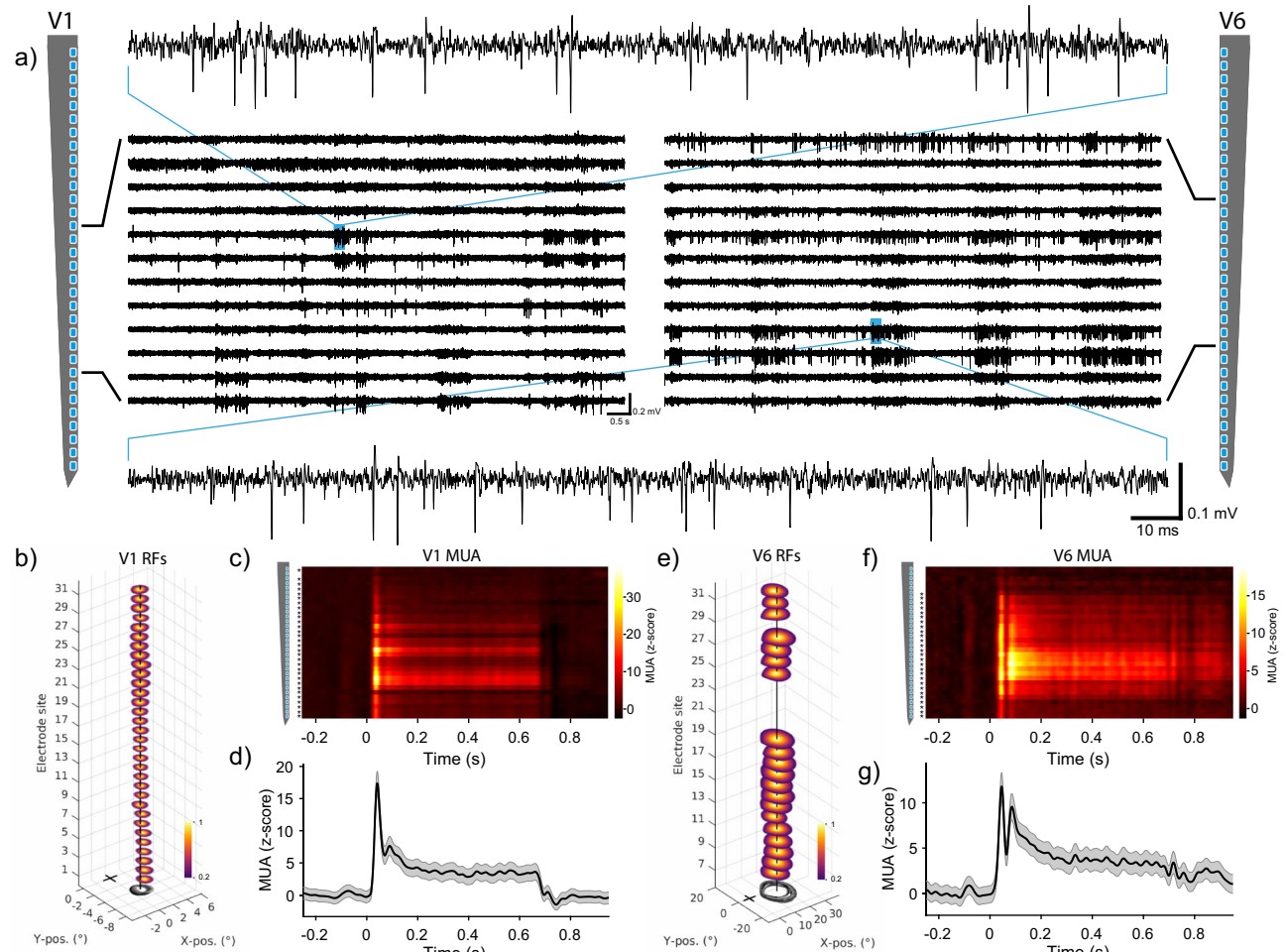

**Fig. 5 | Neural recordings in areas V1 and V6. a** Band-pass-filtered signal (0.3–6 kHz) from example recording sites across one shank in area V1 (left) and area V6 (right). The top and bottom traces show a magnified view of the respective example signals in V1 and V6. **b** Receptive field (RF) locations calculated from the normalized multi-unit-activity (MUA) of all significantly modulated sites along the example shank ($n = 32$ out of 32 sites). Outlines of RFs are shown at the bottom in black-to-gray lines from most superficial to the deepest channel. The vertical black line indicates the median RF location across all sites. The black cross marks the position of the fixation point at the center of the monitor. **c** Trial-averaged MUA along the example shank around the time of visual stimulation with gratings.

Asterisks on the left indicate significant modulation between pre-stimulation baseline (−0.25–0 s) and poststimulus time (0–0.65 s) ($P < 0.05$; two-sided Kolmogorov–Smirnov test for channels with MUA > 3σ). **d** Average MUA ± SEM across all significantly modulated sites from the example V1 shank ($n = 31$ out of 32 sites). **e–g** Same as **b–d** but for example shank in area V6 ($n = 20$ out of 32 sites were modulated during RF mapping; $n = 27$ out of 32 sites were modulated during visual stimulation with gratings). MUA was smoothed with a Gaussian window (σ = 8 ms). Note different axis scaling and RF sizes between area V1 and V6 in panels **b** and **e**. Data for RF mapping and visual stimulation with gratings were recorded in separate sessions in Monkey A. Source data are provided as a Source Data file.

channels with MUA > 3σ, see Methods for details). Figure 5d illustrates the MUA averaged over all modulated sites from an example shank in V1 ($n = 31$ out of 32 sites). Similarly to area V1, many sites in area V6 also showed a significant spatially selective modulation (Fig. 5e) ($n = 20$ out of 32 sites). RFs along the shank mostly overlapped, and many sites exhibited a significant MUA response after visual stimulation with gratings (Fig. 5f, g; $n = 27$ out of 32 sites).

Having established the overall responsiveness and visual selectivity of MUA, we next sorted spiking data into single units. Spike sorting was performed semi-automatically with the "Kilosort" algorithm[31]. Figure 6 depicts, in the left panel of each column, the average waveform across all 32 channels of the relevant electrode shank. Due to the fine inter-electrode spacing (25 μm), spike waveforms of each identified neuron were detectable as a spatial (and temporal) pattern across multiple sites. Raster plots and corresponding peristimulus time histograms (PSTHs) around the time of visual stimulation (black bar on top, 0.65 s duration) can be seen in the first and second row of Fig. 6. The inset in the second row shows orientation tuning curves calculated from the average spiking activity during

the stimulus period (0–0.65 s). A von Mises function was fit to the mean firing rates for visualization[32]. Peak-normalized auto-correlograms for all spikes during the recording session are shown in the third row. The bottom row shows each unit's firing rate over the course of a recording session, documenting that all units were stable throughout the session.

The observed single units exhibited different response characteristics, as expected from neural recordings in the visual cortex. Examples in Fig. 6 were selected in order to depict the variety of response profiles present in the data. The units in Fig. 6a, b were recorded in area V1. Unit (a) was strongly visually driven, showed a sharp peak after stimulus onset and exhibited clear orientation tuning, reminiscent of the principal cells in V1 of the anesthetized marmoset[33]. The unit in Fig. 6b was suppressed during the time of visual stimulation, had a relatively high baseline firing rate, and was orientation tuned. Unit (c) and (d) are examples recorded in area V6. Unit (c) showed a sustained activation and orientation tuning, similar to previous reports in V6[34]. In contrast, unit (d) responded only transiently and exhibited only weak orientation tuning,

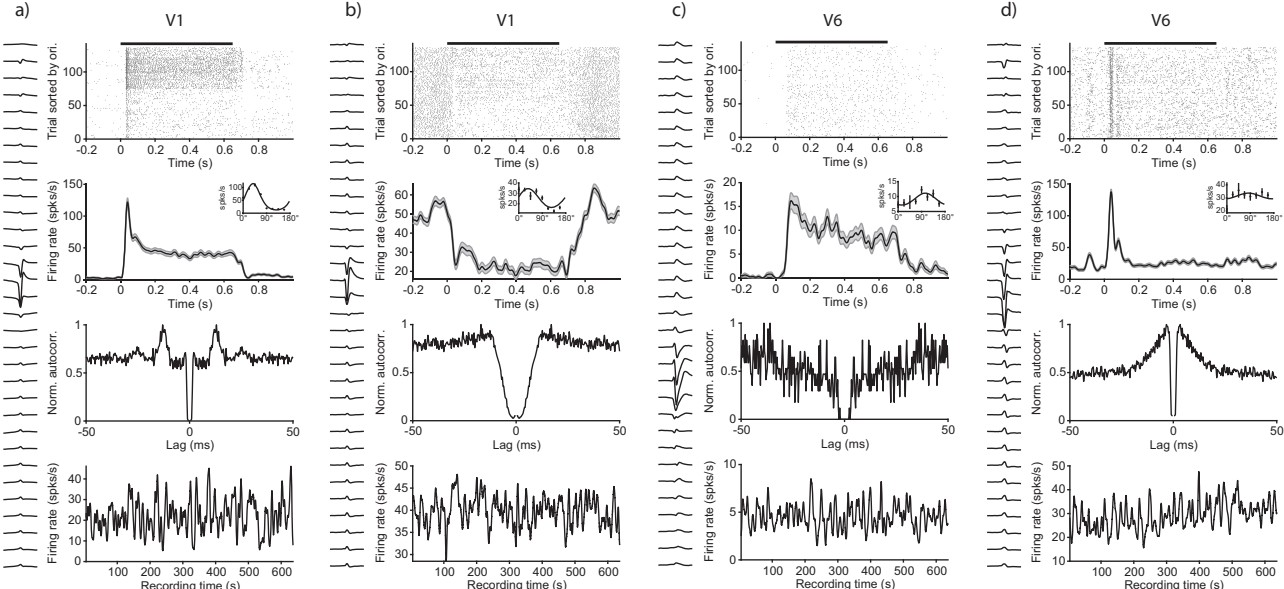

**Fig. 6 | Single-unit examples from areas V1 and V6.** Examples of four visually modulated single units (**a–d**). The left side of each column shows the mean waveform across all 32 recording sites of the electrode shank on which the largest absolute amplitude was detected. Top row: spiking raster plot around the time of visual stimulation. Trials are sorted by orientation condition. Black bar on top indicates stimulus duration (0.65 s). Second row: trial-averaged and smoothed (Gaussian window, σ = 10 ms) peristimulus time histogram (PSTH). Inset shows orientation tuning curves calculated from the mean activity during the stimulus period (0–0.65 s). Error bars and shaded areas indicate SEM. Third row: peak-normalized auto-correlogram for all spikes across the recording. Bottom row: smoothed firing rate (Gaussian window, σ = 2 s) across the entire session, indicating the stability of the recordings. The first and last 5 seconds of data were excluded due to artifacts arising from the connection or disconnection of the recording system. All examples from one recording session in Monkey A. Source data are provided as a Source Data file.

potentially due to a non-optimal spatial frequency of the visual stimulus.

## Optogenetic stimulation of area V6

To demonstrate that our recording approach is compatible with optogenetic stimulation techniques, we injected an adeno-associated viral vector (AAV), expressing the fast channelrhodopsin variant "Chronos"[35] under control of the CamKIIα promotor into area V6. The expression under the CamKIIα promotor is almost exclusively restricted to excitatory neurons[36–38]. After several weeks of expression, we placed an optic fiber above the V6 craniotomy to stimulate neurons underneath the transparent silicone gel (Fig. 1c). The optic fiber was coupled to a laser that could be directly modulated with arbitrary waveforms. Stimulation was performed with sinusoidal waveforms at a peak amplitude of 25 mW. One example stimulation trial is depicted in Fig. 7a.

Optogenetically induced spiking was visible across several channels (Fig. 7a). Analysis of the trial-averaged MUA revealed clear optogenetic activation time-locked to the laser waveform, for all 32 channels along the example shank from Monkey A (Fig. 7b). The z-scored MUA averaged across all trials and all modulated channels is presented in Fig. 7c ($P < 0.05$, Kolmogorov–Smirnov test for channels with MUA > 3σ, $n = 32$ out of 32 channels). Comparable data from Monkey D is shown in Supplementary Fig. 4.

In order to exclude potential contamination from light-induced artifacts, we took several precautions and applied appropriate controls: First, the silicon probes used in this study are relatively robust against light artifacts[39]. Furthermore, we avoided fast transients in light intensity by stimulating with low-frequency sine waves that do not contain energy in the spike frequency range. Data for MUA and SUA analysis in which optogenetic stimulation was performed, were high-pass filtered with a sharp frequency cutoff (Chebyshev Type II filter) and strong stopband attenuation (200 dB) to remove any potential contamination from the low-frequency laser signal[40]. In addition, we

included a control condition, in which light with a wavelength of 594 nm with matched output power was used for optical stimulation. The opsin variant used in this study is not activated by this wavelength[35]. These controls ruled out that the observed neural activation was caused by light artifacts or other non-specific effects such as heating.

Next, we spike-sorted the data as described earlier in order to identify optogenetically modulated single units. Four example units are depicted in Fig. 7d–g (figure conventions are as in Fig. 6). Figure 7d, e shows examples from Monkey A, in which optogenetic stimulation was performed with an 80 Hz sinusoidal pattern. On each trial, sinusoidal waveforms started smoothly at the trough from an intensity of 0 mW with a peak amplitude of 25 mW. Single-unit spikes were precisely time-locked to the laser stimulation (Fig. 7d, e). Consistent with the trial-averaged optogenetic responses, the resulting autocorrelation analysis of SUA showed a prominent peak at the reciprocal of the stimulation frequency (1/80 Hz = 12.5 ms). Figure 7f, g shows examples from Monkey D, in which optogenetic stimulation was performed with sine waves of different frequencies (0, 10, 20, 30, 40, 50, 60, 70, 80 Hz) and randomized phases. To avoid artifacts from sharp transients in light intensity, onset and offset of the stimulation waveform were smoothed (see "Methods" for details). The resulting average laser intensity across all trials is shown on top of the raster plot. In both monkeys, spiking activity of single units was not affected by the control stimulation (yellow and gray traces in the second row, see also Supplementary Fig. 4), and the rates remained relatively stable throughout the recording session (Fig. 7, lowermost row). Ten out of the 21 (≈48%) well-isolated single neurons from the two example recording sessions in the two monkeys were significantly modulated ($P < 0.05$, Wilcoxon rank-sum test for neurons with SUA > 3σ over baseline, Supplementary Fig. 5a). Most of the recorded neurons clustered around the center of the electrode array. No obvious depth pattern with regard to optogenetically induced response strength was observed (Supplementary Fig. 5b).

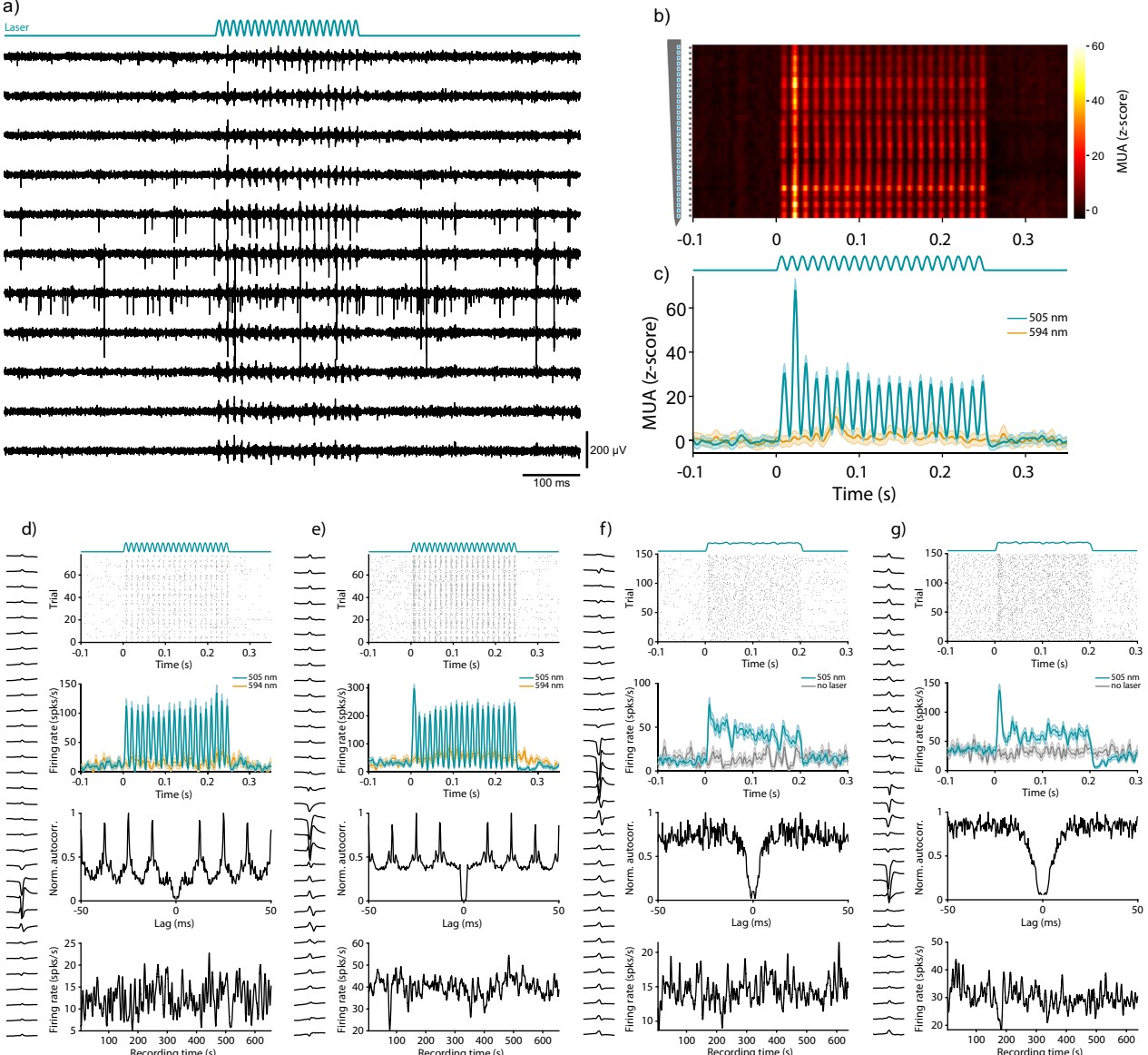

**Fig. 7 | Optogenetic activation of neurons in the awake marmoset. a** Example traces of band-pass-filtered data during optogenetic stimulation with an 80 Hz sinusoidal pattern of 250 ms duration (25 mW peak). **b** Trial-averaged, z-scored MUA of all recording sites from an example shank for the 505 nm stimulation condition. Asterisks on the left indicate significant modulation ($P < 0.05$, two-sided Kolmogorov–Smirnov test for channels with MUA > 3σ) between pre-stimulation baseline window (−25 to 0 ms) and the stimulus window (0 to 25 ms after laser onset; the stimulus window was chosen to be short in order to exclude responses to visual stimuli present in some trials). **c** Average MUA ± SEM across all significantly modulated channels ($n = 32$ out of 32 channels), for stimulation with 505 nm and 594 nm, respectively as indicated by the color legend. Note that the 594 nm condition included a subset of trials with visual stimulation, which explains the transiently enhanced MUA around 70 ms post laser onset. **d**–**g** Four examples of optogenetically modulated single units: **d**, **e** from Monkey A; **f** and **g** from Monkey

D (the latter receiving 200-ms-long optogenetic stimulation). The left side of each column shows the mean waveform across all 32 recording sites of the relevant electrode shank. Top row: raster plot of spikes around the time of stimulation with 505 nm. Average laser waveform across all trials is shown on top. Second row: trial-averaged and smoothed (Gaussian window, σ = 2 ms) peristimulus time histogram (PSTH) for stimulation conditions with 505 nm and control trials (594 nm or no laser). Shaded area indicates SEM. Third row: peak-normalized auto-correlogram for all detected spikes during the recording session. Note the clear optogenetically induced rhythmicity in the autocorrelations of neurons in (**d**, **e**). Bottom row: smoothed firing rate (Gaussian window, σ = 2 s) across the entire session, indicating stability of the recordings. The first and last 5 s of data were excluded due to artifacts arising from the connection or disconnection of the recording system. Source data are provided as a Source Data file.

## Behavioral report of optogenetic stimulation

In order to test whether activation of excitatory neurons in area V6 could be behaviorally reported, we trained one animal (Monkey A) in a visual and optogenetic detection task (Fig. 8a). The animal was required to briefly maintain fixation (100–150 ms) on a central fixation point. After this period, a background stimulus (full-screen circular grating) was presented. After an additional 150–320 ms, a moving visual target with either low or high contrast was presented for 250 ms.

The center of the movement path was placed within the RF of recording sites with clear optogenetic modulation (see "Methods"). Half of these target trials were randomly paired with optogenetic stimulation (250 ms square pulse, 25 mW amplitude, same onset time as the visual stimulus, see Methods for details). An additional condition was included in which optogenetic stimulation was performed in the absence of a visual target. The monkey was rewarded for making a saccade away from the fixation point within 500 ms from the onset

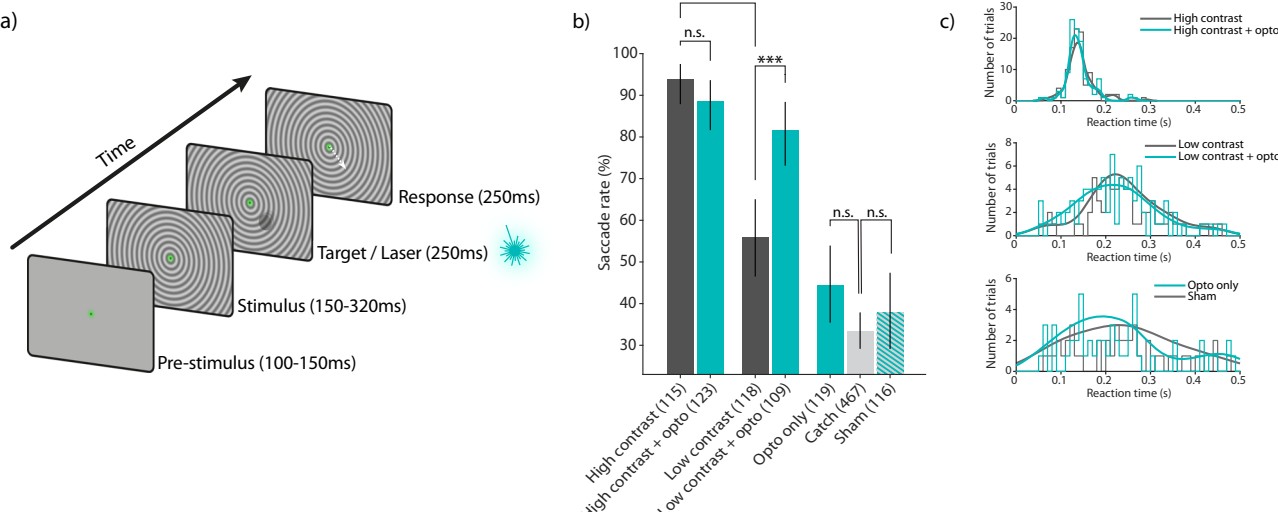

**Fig. 8 | Visual and optogenetic detection task and behavioral results.**
**a** Schematic illustration of the detection task. After a brief pre-stimulus fixation period, a background stimulus was shown, followed by the onset of a small visual target with either high or low contrast. On 50% of these trials, the visual target was paired with optogenetic stimulation. Additionally, trials without visual target were included, either with effective laser stimulation ("Opto only" condition) or with control laser stimulation, in which the optic fiber was placed outside the craniotomy ("Sham" condition), or with no laser stimulation ("Catch"). All trial conditions except catch trials had identical timing and were rewarded if the monkey executed a saccade 50–500 ms after target or laser onset. **b** Saccade rates for all task conditions. The animal showed increased detection performance (higher saccade rate) for high-contrast visual targets compared to low-contrast targets (93.9% vs. 55.9%, $n = 115$ and $n = 118$ trials, respectively; two-sided Chi-squared test; $P = 3.64e\text{-}10$). Pairing high-contrast visual targets with optogenetic stimulation did not result in a difference in saccade rate ($n = 115$ and $n = 123$ trials, respectively; two-sided Chi-squared test; $P = 0.283$). Saccade rate increased significantly when low-

contrast targets were paired with optogenetic stimulation (55.9% vs. 81.7%; $n = 118$ and $n = 109$ trials, respectively; two-sided Chi-squared test; $P = 1.47e\text{-}04$). Optogenetic stimulation alone was not sufficient to be detected by the animal when compared to the false-alarm rate (44.5% vs. 33.4%; $n = 119$ and $n = 467$ trials, respectively; two-sided Chi-squared test; $P = 0.0521$). The saccade rate in the sham stimulation control condition (laser fiber positioned 2 mm outside the craniotomy) was not different from the false-alarm rate (37.9% vs. 33.4%; $n = 116$ and $n = 467$ trials, respectively; two-sided Chi-squared test; $P = 0.419$). The number of total trials per condition (hits+misses or correct rejections + false alarms) are shown in parenthesis. Error bars indicate 95% binomial confidence intervals of the saccade rates. **c** Reaction time (RT) distributions from trials with and without optogenetic stimulation for high contrast (top), low contrast (middle) and in the absence of visual stimulation (bottom). No significant differences in RTs were observed (two-sided Wilcoxon rank-sum test; $P > 0.05$). Smooth lines indicate probability density estimates. Source data are provided as a Source Data file.

time of visual and/or optogenetic stimulation. To prevent false alarms, 40% of all trials were "catch trials", in which neither an optogenetic nor a visual target appeared. In these trials, the monkey was rewarded for maintaining fixation for 800 ms. In order to avoid overestimation of the false-alarm rate due to the longer catch trial duration relative to other trials, we applied a resampling procedure to calculate the true false-alarm rate (see "Methods"). As a control, we randomly interleaved trials with a sham stimulation condition. Sham stimulation was identical to real optogenetic stimulation (without a visual target), but the laser output was switched to a second optic fiber that was placed 2 mm outside the craniotomy. Importantly, sham trials were rewarded identical to real trials, such that the monkey would be able to benefit from any cues unspecific to the optogenetic stimulation.

High-contrast visual targets were correctly reported in 93.9% of trials, compared to only 55.9% in the low-contrast target condition (Chi-squared test; $P = 3.64e\text{-}10$; $n = 115$ high-contrast and $n = 123$ low-contrast trials). Pairing the high-contrast visual target with optogenetic stimulation did not significantly affect detection performance (Chi-squared test; $P = 0.283$; $n = 115$ high-contrast and $n = 123$ high-contrast + opto trials). A different pattern was observed for low-contrast visual targets: Pairing the visual stimulus with optogenetic stimulation caused performance to substantially improve from 55.9% to 81.7% (Chi-squared test; $P = 1.47e\text{-}04$; $n = 118$ low-contrast and $n = 109$ low-contrast + opto trials). The observed increase in saccade rate indicates that the monkey was able to integrate neuronal signals from both, optogenetic and visual sources, in order to improve detection performance. The response rate to catch trials, i.e., the "false alarm rate" was low (33.4%). Interestingly, saccade rates for optogenetic stimulation alone as compared to the false-alarm rate did not quite achieve

statistical significance (Chi-squared test; $P = 0.0521$; $n = 119$ opto only and $n = 467$ catch trials). Importantly, the saccade rate in the sham control condition was not different from the false-alarm rate (Chi-squared test; $P = 0.419$; $n = 116$ sham and $n = 467$ catch trials).

We also analyzed the direction and amplitude of the saccades that led to the correct and incorrect responses described above. The analysis revealed that only the contrast of the visual target but not optogenetic stimulation had a significant effect on saccade amplitude (Supplementary Fig. 7a, Wilcoxon rank-sum test; $P = 6.66e\text{-}10$). Saccade direction did not differ between the tested conditions (Supplementary Fig. 7b). Saccades in catch trials were also directed toward the location of the expected target, although no target was shown is these trials. This behavior was expected, given that the center location of the moving target was identical across trials and therefore predictable.

It is possible that adding optogenetic stimulation results in changes in the animal's reaction time (RT). In the presence of high-contrast visual stimuli, additional optogenetic stimulation might shorten RTs, because the optogenetic stimulation bypasses transmission delays. In the presence of low-contrast visual stimuli, additional optogenetic stimulation might shorten the latency until the combined signal crosses a threshold. In the absence of a visual stimulus, optogenetic stimulation might not lead to significant changes in detection rate (see above comparison of saccade rates in opto-only vs. catch), but it might nevertheless affect RT distributions. To assess these possibilities, we computed RT-histograms and directly compared RTs from trials with and without optogenetic stimulation. Optogenetic stimulation did not result in a significant difference in RTs, neither for high-contrast (Wilcoxon rank-sum test; $P = 0.4395$; $n = 108$ non-opto and $n = 109$ opto hit trials), nor for low-contrast visual stimulation

(Wilcoxon rank-sum test; $P = 0.1830$; $n = 66$ non-opto and $n = 89$ opto hit trials). We also compared RTs from "opto only" trials with RTs from sham stimulation. Sham stimulation was timed identical but could not be detected by the animal, thus providing an RT distribution as expected from false alarms with the similar number of trials. We did not observe a significant difference in RTs between these conditions (Wilcoxon rank-sum test; $P = 0.5451$; $n = 53$ opto-only and $n = 44$ sham hit trials). Furthermore, the RT distribution for "opto only" trials did not show a peak at the late edge of the RT window (0.5 s), which otherwise would indicate that the animal was detecting this type of stimulation but with slow RTs.

We also applied signal detection theory (SDT) to the behavioral data to investigate the effect of optogenetic stimulation on sensitivity and response bias (measured as $d'$ and $c$, respectively[41]). We computed $d'$ and $c$ from "low contrast visual" vs. "catch" trials and compared them to the values from "low contrast + opto" vs. "opto only" trials. Therefore, the "signal" distributions required for the SDT framework are given by the "low contrast visual" and by the "low contrast visual + opto" conditions. Respectively, the "noise" distributions are given by the "catch" trials and the "opto only" trials. Thus, a comparison of $d'$ and $c$ between these conditions allowed us to isolate the effect of optogenetic stimulation during low-contrast visual stimulation. Optogenetic stimulation led to an increase in $d'$ (Supplementary Fig. 6a; $P = 0.0367$; bootstrap test), indicating an increase in sensitivity for the detection of paired visual and optogenetic stimulation when compared to visual stimulation alone. Optogenetic stimulation also led to a decrease in $c$ from positive to negative values (Supplementary Fig. 6b; $P \leq 0.0001$; bootstrap test), indicating that the animal changed its bias from a "no" response, i.e., not responding with a saccade ($c > 0$) toward a "yes" response, i.e., responding with a saccade ($c < 0$). Importantly, while there was no significant difference between the false-alarm rate calculated from catch trials and the hit rate from "opto only" trials (Fig. 8b), it should be noted that the conditions were not rewarded identically. Responses to catch trials were considered false alarms and therefore not rewarded, whereas responses to "opto only" trials were considered hits and therefore rewarded. Thus, a shift in response bias toward negative $c$ values might be expected because the monkey could increase the number of hits by doing so.

## Discussion

The approach presented here enables for the first time neural recordings and optogenetic stimulation in combination with behavioral manipulation in the awake-behaving marmoset. We demonstrate the functionality of our design by obtaining multi- and single-unit recordings in two visual areas simultaneously and using optogenetic stimulation to influence the animal's behavior in a detection task.

Our design relies heavily on the use of 3D printing technology. 3D printing allows for rapid design adaptations, requires few mechanical constraints and enables the production of prototypes at low cost and short turnover times[42,43]. These factors make it possible for other researchers to easily modify and improve the design presented here. There are several potential adaptations that could be useful, for example, expansion of the chamber and change in its position relative to the skull. Such modifications could enable recordings from more lateral brain areas such as area MT or IT, which are inaccessible with the current design (Supplementary Fig. 2). Moreover, the design could be adapted such that it integrates a mechanism for head fixation on the chamber[28,44]. This would make a separate headpost obsolete and thereby allow better access to frontal regions. The integration of a head-fixation mechanism on the chamber might also enhance mechanical stability, which could facilitate the use with imaging techniques. One important drawback of 3D printing methods (specifically sintering methods as used here) is that the untreated surface finish is rough. Therefore, additional steps are required if a high-

precision fit (e.g., for headpost or screw threads) or a watertight sealing is necessary[42].

The weight of the complete implant, allowing recordings from 192 electrodes, amounted to ~8 g (Fig. 2c). The titanium chamber alone weighs only 1.5 g and is designed to smoothly fit onto the surface of the skull with a low profile, thereby minimizing any unnecessary volume (Figs. 1a, b and 2a, b). The achieved weight minimization and the mechanical robustness of 3D-printed titanium makes our design compatible with wireless recording technology. Data loggers with batteries or wireless transmitters might be utilized while remaining at an acceptable weight[12,13,17].

Semi-chronic recording approaches, as presented here, do not require repeated insertions of electrodes into the brain for each recording session. Thus, just like chronic recordings, they can shorten the experimental preparation time and reduce the risk of infections. At the same time, such an approach retains the option of moving probes deeper into the brain after signal decay or in case the recording depth needs to be adjusted[18].

Yet, there are also advantages to other approaches such as chronic or acute recordings. In small animals, e.g., mice, immobile, chronically implanted silicon probes can provide neural recording stability over long periods of time[45–47]. Stability is likely related to the relative absence of movement of the mouse brain inside its skull. In marmosets, recent work has shown good recording stability with chronically implanted floating electrode ("Utah") arrays[17]. However, long-term recording stability with immobile silicon probes remains to be demonstrated. Furthermore, chronically implanted electrode arrays, such as the "Utah" array do not require any movable parts and can therefore be completely sealed off after implantation, minimizing the risk of infections after surgery[16,17]. Acute recording approaches on the other hand allow for repeated independent measurements and can therefore result in higher single-unit yield and make it possible to quickly change recording position[48]. Thus, while semi-chronic recordings are advantageous in many circumstances, individual experimental requirements should be considered when evaluating different recording approaches.

In this work, we performed semi-chronic recordings with silicon probe technology from passive electrodes. Yet, our design is compatible with active probes such as Neuropixels[47,49] in chronic[46,47] or semi-chronic[50] configuration. Currently, electrode shanks and microdrive-mountable components of passive silicone probes as used in this work are still smaller than those of Neuropixels probes (shank width: 25–50 μm vs. 70 μm for Neuropixels; shank thickness: 15 μm vs 20 μm for Neuropixels). However, active probes with fully integrated electronics and miniaturized head stages would allow for even higher channel-count recordings and will be an important next step for the advancement of neural recordings in awake marmosets.

We demonstrated the utility of our design by behavioral manipulation via optogenetic stimulation of area V6 in the context of a detection task. Previous work in macaques has demonstrated that optogenetic stimulation of the primary visual cortex can be readily reported via saccades[51,52]. These findings are consistent with the view that animals perceived optogenetically induced phosphenes. In contrast, our own results from area V6 indicate that optogenetic stimulation alone was not sufficient to significantly modulate saccade rates, despite being close to the statistical threshold ($P = 0.0521$). However, a clear behavioral effect was observed when laser stimulation was paired with a low-contrast visual stimulus. It is known from microstimulation experiments in macaque V1 that detection sensitivity can substantially increase with behavioral training[53]. Furthermore, the detection of microstimulation outside of primary sensory areas can require extended training[54]. Similar changes in sensitivity thresholds have been reported for optogenetic stimulation in the somatosensory cortex[55]. Therefore, it is plausible that further behavioral training in the

marmoset would also lead to a report of optogenetic stimulation alone. This aspect should be investigated in future work.

# Methods

## Animals

All animal experiments were approved by the responsible government office (Regierungspräsidium Darmstadt) in accordance with the German law for the protection of animals and the "European Union's Directive 2010/63/EU".

Five adult male marmosets were implanted with chamber, headpost, and reference wires. Three of these animals were subsequently injected with a viral vector in area V6, and implanted with electrodes in areas V1 and V6. The decision to use male animals was due to availability and was not part of the experimental design. Table 1 lists relevant details, procedures, and outcomes for each animal. All animals were obtained from the German Primate Center (Göttingen, Germany).

## Stimulus presentation

Stimulus presentation was controlled by the custom-developed ARCADE toolbox (https://github.com/esi-neuroscience/ARCADE), based on MATLAB 2014a (Mathworks, USA) and C++. Stimuli were displayed on a TFT monitor (Samsung SyncMaster 2233RZ) at a refresh rate of 120 Hz. Animals were placed at a distance of 45 cm to the monitor in a dimly lit recording booth. A photodiode was placed in the top left corner of the monitor in order to determine exact stimulus-onset times.

## Eye tracking

The left eye of the animals was tracked under external infrared light illumination with a sampling rate of 1 kHz (Eyelink 1000, SR Research, Canada). A 25 mm/F1.4 lens was used at a distance of 28 cm to the animal's eye.

## Implant design

Designs were developed in Blender 2.79 (https://www.blender.org/), OnShape 1.79 (https://www.onshape.com/), and Solidworks 2018 (https://www.solidworks.com/). CT segmentation was performed in 3D Slicer 4.10.0 (https://www.slicer.org/). The skull template shown in Figs. 1 and 2 was based on high-resolution CT data from a marmoset skull archived on the MorphoSource database (https://doi.org/10.17602/M2/M5203). The same skull template was used as anatomical reference for the curvature at the bottom of the chamber.

## 3D printing and implant fabrication

Chambers and microdrive stabilizers were printed via direct metal laser sintering (DMLS) from grade 5 (Ti6Al4V) titanium (Materialise, Belgium). DMLS can produce parts with mechanical and chemical properties comparable to classically CNC-machined titanium. However, the minimum feature size is typically 0.4 mm, and the minimum wall thickness is 0.5 mm. This means that very small corners and sharp edges cannot be printed accurately. Thus, the four screw threads (M1.4 thread diameter, 2-mm screw length) that are used to secure the lid to the chamber were manually added after 3D printing. This was done by either using a handheld tapping drill bit or placing the chamber in the CNC mill to create a thread after manual alignment[42]. Furthermore, due to the sintering process, the surface finish of DMLS parts is rough. Thus, to ensure watertight sealing of the closed chamber, a thin layer of silicone (Kwik-Sil, World Precision Instruments, USA) was applied to the small ridge inside the lid that served as a contact area between chamber and lid.

Both, the headpost as well as its holder (Fig. 1d) were produced by standard CNC milling. 3D printing was not viable here, because it does not offer the precision necessary for the fit between headpost and its holder, without substantial post-processing[42]. All lids were printed via selective laser sintering from PA12 nylon (Shapeways, USA). Nylon was

chosen because of its high abrasion resistance. The use of 3D-printed lids makes it possible to rapidly and flexibly produce multiple versions of lids. Before electrode implantation, the inside of the chamber does not contain any parts other than the (optional) reference wires. Therefore, the initial version of the lid was flat and could later be replaced by a taller version. This procedure allowed the animals to gradually get habituated to the size and weight of the final implant. All custom implantation holders and guides were printed from standard resins via stereolithography on a "Form 1" printer (Formlabs Inc., USA).

## CT scans, segmentation, and alignment

CTs were performed with a Planmeca ProMax 3D Mid scanner (Planmeca Oy, Finland) at 90 kV and 10 mA with a voxel size of 150 μm (isotropic) under brief anesthesia induced with an intramuscular (i.m.) injection of a mixture of alfaxalone (8.75 mg/kg) and diazepam (0.625 mg/kg). The anesthetized animal was placed on a small adjustable bed on which a heating pad was mounted. A plastic headpost holder was then used to secure the animal's head in position for the duration of the scan. During CT imaging, animals were not aligned in stereotaxic coordinates. Instead, the chamber and screws were used as fiducial markers for post-scan alignment, as described below. After the scan was completed, CT data were loaded into the 3D Slicer software for segmentation, i.e., delineation of regions in the images that correspond to metal parts, cement and bony tissue. Segmentation was performed by simple intensity thresholding ("Threshold" function in the segment editor of 3D Slicer). Threshold values for upper and lower cutoff were manually set for each animal such that the desired regions of cement, metal or bone were clearly outlined. Specifically, the thresholds for the bony tissue (semi-transparent gray in Figs. 3e–g and 4e–g) was adjusted until the intracranial space was clearly outlined, such that it could be later used to fit the 3D template brain for coordinate panning. The thresholds for the segmentation of the chamber and cement were adjusted until the upper rim of the chamber was clearly outlined, such that the segmented region could later be aligned with the 3D model of the chamber. This was important because the chamber served as the reference position for the implantation targets in the brain (see also 3D-printed guide in Supplementary Fig. 3). A third threshold-based segmentation was performed that made the four stainless steel screws on the side of the chamber visible. The positions of the four screws were used as additional fiducial markers for alignment to the model of the chamber. All threshold values are listed in Supplementary Table 1. Segmented volumes were then exported as STL files and imported into Blender for alignment. All CT-based imported volumes were aligned to the 3D model of the chamber. Alignment was performed by manual translation and rotation such that the position of the screws and the upper rim of the chamber in the segmented data aligned with the corresponding positions in the chamber model. Because the chambers had been implanted with the animal aligned in the stereotaxic frame, this effectively brought the CT segmentation data back into stereotaxic coordinates.

## Planning of implantation targets

In Monkey A, coordinates for the implantation targets were based on ref. [56]. In monkeys D and U, coordinates were based on the following procedure: First, we loaded the MRI-based template marmoset brain segmentation from ref. [29] into Blender. Specifically, we loaded the segmented volumes of the whole brain (red color in Figs. 3e–g and 4e–g), and areas V1 and V6 (DM). The MRI data and segmentation can be downloaded at https://marmosetbrainmapping.org/download_atlasv1.html. The template brain (together with area delineations for V1 and V6) was then transformed to fit exactly into the intracranial space of the segmented CT data of each animal. This process was performed manually by translating and scaling in all three spatial dimensions, and rotating in the pitch axis. For better visualization during alignment, we used the "clipping border" function in Blender

**Table 2 | Coordinates of implantation targets**

|  | V1 caudal from interaural line | V1 lateral from midline | V6 caudal from interaural line | V6 lateral from midline |
|---|---|---|---|---|
| Monkey D: | 7.7 mm | 1.3 mm | 2.6 mm | 4.1 mm |
| Monkey U: | 7.0 mm | 1.3 mm | 2.2 mm | 3.9 mm |
| Monkey A: | 8.5 mm | 1.3 mm | 2.5 mm | 3 mm |

which allows viewing coronal, sagittal, and horizontal slices of the segmented volumes. Furthermore, for alignment and later visualization, the segmented volume of the skull was set to an alpha value of 0.2 to appear semi-transparent.

This alignment procedure determined where the expected locations of areas V1 and V6 were, relative to the animal's skull and to the already implanted chamber. Thus, for each animal and each area, we selected a target location for implantation based on the individually fitted template brain. For V1, we selected a location close to the midline and close to the border to V2 because this region is known to represent visual space in the lower visual field close to the vertical meridian at an eccentricity of -3–7° of visual angle[2]. For V6, the retinotopic map is more complex. Nevertheless, we aimed at targeting the part of V6 with similar, intermediate eccentricities based on the maps presented in ref. [57]. From the resulting target coordinates for each animal (Table 2), we created a 3D-printed implantation guide (stencil) for each animal (Supplementary Fig. 3). In the second surgery, the guide was temporarily placed on the chamber and the two holes pointing at the target locations could be used to mark the positions for the craniotomies in areas V1 and V6.

### 3D rendering
3D renderings were generated via viewport rendering in Blender ("Blender Render" setting) and exported via screen capture with Matlab (code and Blender files are available online, see "Data availability"). For anti-aliasing, the OpenGL multi-sampling setting in Blender was set to a value of 16. For visualization of the implantation sequences (Figs. 3a–d and 4a–d), we used Blender's "key frames" feature.

### Anesthesia
Anesthesia for all surgeries was induced with an intramuscular (i.m.) injection of a mixture of alfaxalone (8.75 mg/kg) and diazepam (0.625 mg/kg). Tramadol (1.5 mg/kg) and metamizol (80 mg/kg) were injected i.m. for initial analgesic coverage. Subsequently, a continuous intravenous (i.v.) infusion was provided through the lateral tail vein. The i.v. mixture contained glucose, amino acids (Aminomix 1 Novum, Fresenius Kabi, Germany), dexamethasone (0.2–0.4 mg kg$^{-1}$ h$^{-1}$), tramadol (0.5–1.0 mg kg$^{-1}$ h$^{-1}$) and metamizol (20–40 mg kg$^{-1}$ h$^{-1}$). The maximal infusion rate was 5 ml kg$^{-1}$ h$^{-1}$. Animals were breathing spontaneously throughout the surgery via a custom 3D-printed face mask that applied isoflurane (0.5–2% in 100% oxygen). Heart rate, respiration rate and body temperature were constantly monitored (Model 1030 Monitoring Gating System, SAII, USA).

### Implantation of chamber and headpost
After placing the animal in a stereotaxic apparatus for the first surgery, an incision was made on the dorsal part of the skull. The temporal muscle was slightly retracted (<5 mm from the superior temporal lines) and all soft tissue was completely removed from the bone surface. No resection of the muscle was necessary. Marmosets have thin skulls and a narrow subdural space, which can make the use of bone screws problematic. Therefore, we used only dental adhesive and cement to secure the implant to the skull[28]. The bone was first cleaned by mechanical abrasion, then scrubbed with 5% $H_2O_2$ and rinsed with saline. To ensure strong attachment of the implant to the skull, we

recommend to prepare an area of at least 15 × 25 mm (Fig. 2b) such that enough surface area is available on which the cement can bond to the skull. For an optimal bonding between cement and bone, the skull surface was roughened with a metal brush, and any remaining dust was removed. After the bone was completely clean and dry, we applied a thin layer of light-curable dental adhesive (All-Bond Universal, BISCO). After drying and curing with blue light, we applied a thin layer (<1 mm) of dental cement on top of the adhesive. Once the cement was cured, a small bur hole was drilled just anterior of the chamber. Two platinum wires (PT-5T, Science Products) were implanted epidurally at this location and served as backup reference wires for the recordings (the actual reference wires were later implanted subdurally in the second surgery).

### Injection of the viral vector
Viral vectors were injected with a microinjector pump (UMP3-1, World Precision Instruments), holding a 10 μL microsyringe (NanoFil syringe, World Precision Instruments) to which a 35 G injection needle was attached. A durotomy of -1.5 mm was performed with a bent 25 G cannula, and the vector was injected at two depths (−1.4 mm and −0.5 mm from the surface). A volume of 2.5 μl at each depth was injected at a speed of 200 nL/min (total injected volume = 5.0 μl). To ensure sufficient diffusion of the viral vector, we waited 10 min after each injection before moving or retracting the needle. The viral vector used in this study (AAV1.CamKIIa.Chronos-eYFP-WPRE) was obtained from Vector Biolabs (USA).

### Silicon probes
Silicon probes were semi-chronically implanted in areas V1 and V6, mounted on one microdrive per area (Nano-Drive CN-01 V1, Cambridge NeuroTech, UK). Two 32-channel shanks with 250 μm spacing were implanted in V1, and four 32-channel shanks in V6 (H2 probe, Cambridge NeuroTech, UK). Electrode implantation was performed directly following the injection of the viral vector. Electrode tips were disinfected shortly before the implantation by dipping them twice in 70% ethanol for 45 s. After the electrodes were in place and the cement was hardened, craniotomies were sealed by applying several drops of soft silicone gel (DOWSIL 3-4680, Dow Corning).

### Acquisition and processing of neural data
Neural signals were recorded with the OpenEx suite 2.2 software package (Tucker Davis Technologies, USA) through active, unity gain head stages (ZC32, Tucker Davis Technologies, USA), digitized at 24,414.0625 Hz (PZ2 preamplifier, Tucker Davis Technologies, USA) and re-sampled offline to 25 kHz. Sample-by-sample re-referencing was applied by calculating the median across all channels for each shank and subtracting this signal from each channel of the corresponding shank[49]. Data were band-pass filtered for spiking activity either with a 4th-order Butterworth filter (0.3–6 kHz) or, in case optogenetic stimulation was performed, with a 40th-order Chebyshev Type II filter (0.3–8 kHz) with a stopband attenuation of 200 dB to exclude any contamination from lower frequencies. For further analysis, multi-unit activity (MUA) was calculated by full-wave rectification, filtering with a 6th-order low-pass Chebyshev Type II filter (stopband edge frequency of 500 Hz, stopband attenuation of 50 dB) and downsampling to 1 kHz. All processing and analysis were performed in MATLAB 2020b unless otherwise noted.

### Optogenetic stimulation
Optogenetic stimulation was performed with a laser beam combiner (LightHUB, Omicron laserage), housing a 100 mW diode laser with a wavelength of 505 nm (LuxXplus 505-100) with direct modulation and a 100 mW DPSS laser with a wavelength of 594 nm (OBIS 594-100) with direct modulation. The combined lasers were coupled to a 50 μm/ 0.22NA optic fiber which was connected to a fiber optic cannula

(200 μm core diameter, 0.39 NA, Doric Lenses Inc.). The cannula was held by a micromanipulator (SM-25C, Narishige) and was positioned ~4 mm above the craniotomy during recording/stimulation sessions. Laser power was calibrated prior to the experiments with a photodiode-based optical power meter (PM130D, Thorlabs). Output power was measured at the tip of the fiber optic cannula. Laser waveforms were generated by a real-time signal processor (RZ2 bioamp processor, Tucker Davis Technologies, USA). To avoid artifacts arising from sharp transients in laser intensity[58], we only used smooth on- and offsets[40]. This was done by using one half of a sine wave as a taper at the beginning and end of any sharp signal (5 ms trough-to-peak time, with the trough having an intensity of 0 mW).

## Spike sorting and single-unit analysis

Spike sorting was performed offline with Kilosort2[31]. Average spike waveforms were calculated from the trimmed mean (5% outlier exclusion). Automatically detected clusters from Kilosort2 (putative single units) were excluded if they showed more than 2.5% spiking within the inter-spike-interval time below 1.5 ms (violation of the refractory period). Clusters with biophysically implausible waveform shapes were rejected similar to ref. [59]. For each cluster, the mean waveform from the channel with the largest amplitude was compared to a set of 20 manually pre-selected "good" template waveforms. If the Pearson correlation coefficient between a given mean waveform and any of the template waveforms did not reach 0.95, the cluster was rejected. In addition, we rejected clusters that were not sufficiently localized in space[59]. For this, we determined the number of channels whose waveform amplitude exceeded 50% of the peak amplitude. Only clusters that were concentrated within <5 channels were considered for analysis. Autocorrelation functions were generated at a resolution of 0.33 ms and scaled by dividing by the maximum value after the removal of the central peak.

## Receptive field mapping

All details about the RF mapping procedure have been described previously[30]. RF mapping was performed with stimuli consisting of black wedges and annuli of various orientations and sizes, presented on a gray background for a duration of eight frames (120 Hz monitor refresh rate). For RF calculation, MUA data were cut into epochs of 280 ms (from 100 ms before to 180 ms after stimulus onset). Epochs were included in the analysis if the eye position remained inside the fixation window throughout the epoch. For noise-rejection purposes, we excluded epochs in which the standard deviation of MUA across time was more than 10 times larger than the median standard deviation across all epochs. Sites were considered to be modulated if the mean MUA from at least three different wedge stimuli and at least three different annulus stimuli evoked a response that was significantly larger (one-tailed, paired $t$ test; $P < 0.01$) than the MUA during baseline (100 ms to 0 ms prior to stimulus onset). For plotting, MUA was normalized per site to have a value between zero and one. RF plots and outlines were generated by truncating the normalized MUA at a value of 0.2.

## Passive fixation task

A passive fixation task was used to measure neural responses following visual stimulation with gratings (Figs. 5 and 6). At the beginning of each trial, the animal was required to maintain its gaze at a central fixation point within a window of 1.4° radius for 100–120 ms. After this period, a static square-wave grating with a spatial frequency of 2 cycles/° was presented for 650 ms at a Michelson contrast of 80%. The size and orientation of the grating was selected at random for each trial. Possible values for the grating radius (in degrees of visual angle) were: 5°, 7.25°, 9.5°, 11.75°, and 14°. Possible values for the grating orientation were: 22.5°, 45°, 67.5°, 90°, 112.5°, 135°, 157.5°, and 180°. After the stimulus offset, the animal was required to maintain its gaze

in the fixation window for another 100 ms. After a correct trial, a picture of a marmoset face was displayed in the center of the monitor, and the animal was rewarded. The amount of reward was 0.07 ml per trial at the start of the session and linearly increased by 0.02 ml per 10 ml consumed (capped at 0.1 ml per trial). The reward was provided via a lick spout and consisted of diluted *gum arabic*.

## Visual and optogenetic detection task

At the beginning of each trial of the detection task, the animal was required to position its gaze at a central fixation point within a window of 1.5° radius for 100–150 ms. After this period, a background stimulus was presented, while the monkey maintained fixation. The background stimulus was a full-screen circular grating, concentric to the fixation point and either contracting towards or expanding from the fixation point, each in a random half of the trials (contrast = 40%, spatial freq. = 2 cycles/°, temporal freq. = 1 cycle/s). At 150–320 ms after the onset of the background stimulus, a black, moving circular patch (1.8° diameter, moving at 5.74°/s, linear motion, random direction) with either high contrast (50%) or low contrast (7.8%) was presented for 250 ms. The center of the movement path of the circular patch was placed at an eccentricity of 8.98° and a polar angle of −72.7° (lower right quadrant). This position fell within the RFs of recording sites on which strong optogenetic modulation was observed. The position was set approximately to the polar angle of the RFs and at the lower end of the area covered by the respective RFs (Supplementary Fig. 7a). This was done in order to make it easier for the animal to perform saccades (head-fixed marmosets have a limited oculomotor range[60]). RF location and size were determined in a prior recording session from MUA data (see Extended Data Figure 7–1 in ref. [30]). In addition, a condition was included in which only optogenetic stimulation was performed in the absence of a visual target. Furthermore, a control "sham" stimulation condition was included, with sham trials being identical to real optogenetic stimulation trials (without a visual target), but with the laser output switched to a second optic fiber that was placed 2 mm outside the craniotomy. All of these "go" trials (60% of all trials) were categorized as hits if the animal made a saccade away from the fixation point within 500 ms after the onset of the moving circular patch or the laser. Responses that were faster than 50 ms were categorized as early responses and were not rewarded. 50% of trials with a visible target were coupled with optogenetic stimulation that consisted of a 250 ms square pulse with an amplitude of 25 mW. The onset timing for visual and optogenetic stimulation was determined by the computer controlling the visual stimulation. We did not compensate for any delay between trigger onset and actual onset of the visual stimulus on the monitor. In the remaining "catch" trials (40% of all trials), no visual or optogenetic target was presented, and the monkey was rewarded for maintaining its gaze at the fixation point for 800 ms. After a correct saccade, or a correct rejection (maintained fixation), a picture of a marmoset face was displayed in the center of the monitor, and the animal was rewarded. The amount of reward was 0.0625 ml per trial at the start of the session and increased by 0.02 ml for every 10 ml consumed (capped at 0.1 ml per trial).

In the detection task described above, catch trials were longer than the average go-trial. Thus, simply calculating saccade rates from catch trials would lead to an overestimation of the true false-alarm rate, because the monkey had more time to perform a saccade in a catch trial than in a go-trial. False-alarm rate calculation was therefore performed in the following way: One randomly selected catch trial with false alarm was compared with the timing of a randomly selected go-trial. If the time of the false alarm from the selected catch trial fell within the time window in which the monkey would have performed a hit (50–500 ms after probe onset), the trial was categorized as a false alarm. If the false alarm timing was such that the monkey would have missed the target, the trial was categorized as correct rejection. This random pairing was performed for $n = 467$ random pairs of trials, as

this was the expected number of catch trials (40% of all trials), given the total number of hits and misses performed by the animal ($n = 700$). The proportion of false alarms and the respective binomial confidence intervals were then calculated for this random sample. This procedure was repeated 1000 times, and the false-alarm rates and confidence intervals from all shuffling iterations were averaged.

### Saccade analysis

Eye data was first smoothed with a Gaussian window of 3 ms STD. Saccades were then detected based on a velocity threshold of 50°/s. During the task, the response of the animal was detected when its gaze position left the central fixation window. Thus, correct or incorrect trials could in some cases result from slow drifts or eye blinks. We, therefore, excluded 11 trials from the saccade analysis in which no saccade endpoint could be determined. For the remaining trials, saccade endpoints were calculated based on the median eye position of the 25 ms interval after the end of a saccade.

### Signal detection theory analysis

Sensitivity ($d'$) was defined as the distance between the signal and the noise means in standard deviation units[41] and calculated from the hit rates and false alarm rates:

$$d' = Z(HitRate) - Z(FalseAlarmRate) \tag{1}$$

Where $Z()$ is the inverse cumulative distribution function of the Gaussian distribution.

Response bias ($c$) was defined as the distance between the criterion and the neutral point where the noise and signal distributions cross over, i.e., where neither response is favored:

$$c = -\frac{1}{2}(Z(HitRate) + Z(FalseAlarmRate)) \tag{2}$$

Confidence intervals for $d'$ and $c$ were calculated by a bootstrap procedure across trials (hits, misses, correct rejections, and false alarms) with 10,000 bootstrap replications. For each replication, a vector of ones and zeros was generated by sampling with replacement from the original samples of hits and misses (or false alarms and correct rejections, respectively) for each condition. These vectors were then used to calculate $d'$ and $c$ 10,000 times. Thus, the confidence intervals are given by the 5th and 95th percentiles of the resulting distributions. Statistical significance of the differences in $d'$ or $c$ values between different conditions was assessed via the calculation of $P$ values from bootstrapping[61] with 10,000 replications. For each replication, the difference of the $d'$ values between conditions, and the difference of the $c$ values between conditions was calculated. The $P$ values were calculated by finding the proportion of values in the distribution of differences that were equal to or larger than the observed differences between conditions in the sampled data. The smallest possible $P$ value for 10,000 replications is therefore 0.0001.

### Statistical analysis

MUA channels were considered to be modulated by visual (Fig. 5) or optogenetic (Fig. 7) stimulation if they fulfilled both of the following criteria: (1) The absolute magnitude of trial-averaged MUA exceeded the value of 3 STDs over the baseline (|z-score| >3) and (2) the distribution of MUA values was significantly different ($P < 0.05$, two-sided Kolmogorov–Smirnov test) between baseline and stimulus period. For testing MUA responses to visual stimulation with gratings, baseline and stimulus periods were defined as −0.25 to 0 s and 0 to 0.65 s from stimulus onset, respectively. For testing MUA responses to optogenetic stimulation, baseline and stimulus periods were defined as −25 to 0 ms and 0 to 25 ms from stimulus onset, respectively.

Single units in the optogenetic population analysis (Supplementary Fig. 5) were considered significantly modulated if (1) the trial-averaged SUA exceeded the value of 3 STDs over the baseline and (2) the distribution of SUA values was significantly different ($P < 0.05$, two-sided Wilcoxon rank-sum test) between stimulation with 505 nm and the control condition. For the control condition in Monkey A, data from trials with 594 nm stimulation and without laser stimulation were pooled. For Monkey D, the control condition included only data without laser stimulation. For testing SUA responses to optogenetic stimulation, baseline and stimulus periods were defined as −0.25 to 0 s and 0 to 0.25 s in Monkey A and −0.20 to 0 s and 0 to 0.20 s in Monkey D, respectively, reflecting the durations of laser stimulation.

Statistical significance between conditions of the behavioral detection task (Fig. 7) was assessed by a pairwise Chi-squared test ($P < 0.05$, $n = 7$ conditions), after multiple comparisons correction via the Benjamini–Hochberg procedure ("pairwise.prop.test" function of the "R" package version 4.0.4).

### Reporting summary

Further information on research design is available in the Nature Portfolio Reporting Summary linked to this article.

## Data availability

The neural and behavioral data generated in this study have been deposited in the Zenodo repository under the accession code https://doi.org/10.5281/zenodo.7259686. Design files for 3D printing and visualization have been deposited in the Zenodo repository under the accession code https://doi.org/10.5281/zenodo.7259721. The CT data used to create the template marmoset skull in this study are available in the MorphoSource database under the accession code https://doi.org/10.17602/M2/M5203. Source data are provided with this paper.

## Code availability

Code to reproduce all figures from this manuscript is available at https://github.com/PJendritza/MultiAreaOptoMarmo/. The version of the code used in this study was archived in the Zenodo repository under the accession code https://doi.org/10.5281/zenodo.7460362.

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

## Acknowledgements

We thank Marianne Hartmann for her constant support in training the animals, Martin Vinck for providing access to his GPU systems, and Gustavo Rohenkohl for his feedback on the manuscript. This work was supported by DFG (SPP 1665 FR2557/1-1, FOR 1847 FR2557/2-1, FR2557/5-1-CORNET, FR2557/6-1-NeuroTMR, FR2557/7-1-DualStreams to P.F.), EU (HEALTH-F2-2008-200728-BrainSynch, FP7-604102-HBP, FP7-600730-Magnetrodes to P.F.), a European Young Investigator Award to P.F., National Institutes of Health (1U54MH091657-WU-Minn-Consortium-HCP to P.F.), the LOEWE program (NeFF to P.F.).

## Author contributions

P.J., F.J.K., and P.F. designed the research; P.J. designed 3D-printable parts. P.J. devised the implantation strategy; P.J. analyzed the data; P.J. and F.J.K. performed experiments; P.J. and P.F. wrote the manuscript. All authors corrected and approved the final manuscript.

## Funding

## Competing interests

The authors declare the following competing interests: P.F. has a patent (applicant: Ernst Strüngmann Institut gGmbH; inventors: Pascal Fries, Christopher Lewis; numbers [and corresponding status]: CA2948406C [active], US11141112B2 [active], JP2017523887A [pending], EP3148454B1 [active], CN106535793B [active]) and is beneficiary of a respective license contract with Blackrock Microsystems LLC (Salt Lake City, UT, USA). P.F. is a member of the Advisory Board of CorTec GmbH (Freiburg, Germany). The remaining authors declare no competing interests.
