## [Peer Review File · Nature Communications]

Multi-area recordings and optogenetics in the awake,
behaving marmosetREVIEWER COMMENTS

Reviewer #1 (Remarks to the Author):

General comments:

The manuscript by Jendritza and colleagues presents a novel methodology for multi-area recordings and optogenetic stimulation in visual cortex of the marmoset monkey with note worthy results. It presents a methodology that will be of high significance in the emerging marmoset animal model. The manuscript is well written and provides excellent illustrations. The methods are sound and exceed standards in the field. They demonstrate the utility of semi-chronic recordings with silicon linear arrays in marmoset visual cortex both in area V1 and V6. They also introduce methods for optogenetic stimulation and demonstrate in a single monkey that viral expression is was sufficient to modulate performance in a visual detection task. I did not find any major flaws in data analysis or interpretation, although I may suggest a couple additional analyses. Overall, the manuscript provides a valuable resource to the field and note worthy findings.

I have two comments for additional analyses to augment assessment of optogenetic stimulation on behavior: 1) Can the authors more directly determine if the behavioral improvements reflect increases in sensitivity or response bias? 2) Can the authors verify if animals might have learned to detect optogenetic stimulation with the current data, but which is only visible when considering longer response intervals than afforded by the timing control (methods line 656-669) that they implement. Beside those two comments, my other comments are minor and largely for clarity.

Major comments:

Figure 8: Is there an increase in sensitivity in the detection task? While false alarms and hit rates are reported which would suggest d-prime increases, but it is not directly computed. It should be possible to directly computing d-prime for (low con stim) vs (catch trial) as compared to the dprime for (low con + opto) vs (opto only) and confirm if sensitivity increases or if this could be a change in threshold? Related to this, at line 461, authors use the term "behavioral effect", but it may be possible clarify in the current data if that is an increase in sensitivity or response bias.

The control for saccade time intervals in computing false alarm rates (for Fig. 8) is interesting, and though reasonable, may deserve brief mention in the main text besides only being mentioned in the methods (lines 656-669), as it is somewhat central to the results presented. In the main text it is reported that "opto only" did not produce more saccades than "catch trials" (marginal effect, $p = 0.0521$); however, this comparison is including a control on timing to match the response intervals to those of go trials with a visual stimulus, correct? This opens a possibility that marmosets did detect the optogenetic stimulation events as compared to catch trials, but made saccades to those opto only events with a slower reaction time than compared to the go trials including a visual stimulus (i.e., the optogenetic event may have been detectable but weaker than a low contrast stimulus). As the authors already note with further training it seems likely marmosets would learn to also detect the optogenetic event (discussion lines 466-468), and possibly if the control for saccade timing is lengthened here, there may already be some evidence of that effect in the current data?

Minor comments:

In all references: it seems inconsistent if volume number and page numbers are reported for citations.

Line 76-78: It was unclear to me how the 3D titanium stabilizers minimize the gap between the bottom of the microdrive and skull? Are the stabilizers screwed into the skull somehow, thus avoiding a gap due to dental cement? This becomes clear, that stabilizers are placed in cement, only later.

Line 78: "Thus making the implantation process easier and faster" this should be a clause, not a sentence?

Line 236-37: How was a recording site on the array determined to be visually modulated is only

mentioned in the Figure 5 caption. It could either be in main text, referenced into methods, or the reference to the figure caption where it is currently described (for the asterisks in 5c) could be more clear in the main text.

Line 254: “Note different scaling between b and e” I think that b and g were intended?

Line 285: The smooth fit curves to orientation tuning are Von Mises or Gaussian fits (include a reference to a paper using the curve, or give details in methods).

Line 279: What was the spatial frequency of the stimulus used in Figure 6? As far as I can tell this was never covered in methods (Passive fixation task, line 615) but apparently the spatial frequency was fixed.

Line 332: Four example units are illustrated in Figure 7 that showed reliable optogenetic activation. What number of neurons in the total population showed similar significant modulations? Was there any specific distribution of which showed stimulation as a function of electrode channel, reflecting any bias for better stimulation at superficial sites possibly? Does light stimulation penetrate to deeper sites?

Line 367: Was the location of the moving target positioned in the V6 neuron’s receptive fields? This is not explicitly stated but I assume it is the case, and it is mentioned in methods but seems relevant to the main text.

Line 390: Typo: “no” should be “not”

Reviewer #2 (Remarks to the Author):

This manuscript describes novel instrumentation and procedures that allow the simultaneous recording of multiple neurons from two areas of the brain of awake, behaving marmosets, with optional optogenetic stimulation. Recordings are shown to be stable and of high quality. Optogenetic stimulation of area V6 influenced behavior in a near-threshold visual saccade task.

None of the results presented in this paper are particularly surprising, but the technology is impressive and will likely be of broad interest to the readership of Nature Communications. The inclusion of raw data (e.g. Figure 5a and 7a) is a further strength. I am enthusiastic about the work and have only a few minor comments:

Line 77: More information should be provided regarding the 3D printed titanium stabilizers and how they interface with the microdrives. Perhaps a figure (main or supplementary) would help.

I was confused regarding surgical implantation of the reference wires. Line 140 indicates that they were implanted during the initial surgery, but line 183 indicates that they were implanted during the second. I think the resolution to this issue is provided in line 540-543, which indicates that two sets of reference wires (backup and actual) were inserted during different surgical procedures. Please clarify.

Figure 3 legend: “Note the cross-shaped markers on the side of the holder...”. An arrow pointing to the small cross-shaped markers would be helpful.

Are the rasters shown in Figure 6 from a single condition or multiple conditions? Does the variability in the spiking response across trials reflect changes in the visual stimulus or other factors?

An analysis of saccade direction and amplitude would bolster the behavioral component of this study. Did saccade metrics vary by condition? On trials in which optogenetic stimulation was delivered without visual stimulation, and in which the monkey made a saccade within the appropriate time

window, were saccades usually directed towards the receptive fields of the stimulated neurons? How about during the catch trials in which the monkey made a false alarm?

The effect of optogenetic stimulation on saccade probability is reported as not significant, but the p-value, 0.0521, exceeded the arbitrary, conventional threshold of 0.0500 only barely. A phrase like “trending toward significance” or “did not quite achieve statistical significance” might better capture the borderlineness of this negative result. This issue reemerges in line 460, which states that “optogenetic stimulation alone was not sufficient to significantly modulate saccade rates”.

Computer aided design materials should be made available.

Reviewer #3 (Remarks to the Author):

This study described a sophisticated and well-described method of preparing marmosets for combined fiber-based optogenetics and multi-electrode electrophysiology. It's a tour de force of methodological engineering in a species that is quite tricky to use. The paper is well written and clear.

Major comment:

The experimental aspects of the study are straightforward and though not themselves groundbreaking, they are excellent and sufficient to establish the success of the method: I have no concerns. It's the hardware development and design, and surgical implantation, that is the critical advance here, especially with respect to the development pathway and implantation of the 3D-printed parts. I felt there was not quite enough detail here before the method to be replicated. I would like to see the authors spell out more fully the entire development cycle pathway for one animal including the step-by-step process of imaging the animal (and segmentation of CT imaging into both hard bony and soft (brain) tissues), the processing and registration of imaging volumes, the rendering choices of blender (and any other software) used for engineering and visualization, including choices made for optimal engineering of the implants as well as the glass brain displays shown in the wonderful figures. The precise choices that must be considered for moving the 3D design to the slicing software and any critical decisions that are required for the metal laser sintering process are key missing details (or at least they are not highlighted well enough for readers to replicate). Besides the adding of threads, which the authors mention is a manual process (which threads and why are they needed, if the implant is fastened to the skull with acrylic cement?), what other post-processing choices are required? A similar sequence of description of the surgical process would also be helpful. For example, it is unclear to me why acrylic cement was chosen over screws to attach the device to the skull. Any caveats to consider with respect to choosing acrylic versus direct hardware attachments to bone?

General remarks to the reviewers

We thank the reviewers for their comments. We believe that the revised manuscript is a substantial improvement over the initially submitted version. We would also like to apologize for the long time it took to finalize the revised manuscript. Personal circumstances of the corresponding author forced us to take more time than anticipated and we also wanted to make sure that our responses could meet the standards expected from the reviewers and the editor.

During the review process, we also performed a code review and corrected two errors:

(1) In Fig. 7b, a subset of trials had been falsely excluded. After correction, the results are qualitatively unchanged.

(2) The labeling of the optogenetic control conditions in the single unit data presented in Fig. 7 was incorrect. Specifically, data from trials without laser stimulation was falsely plotted as '594 nm' control condition. We corrected this such that the yellow lines in Fig. 7d and e now show the correct data from the '594 nm' control condition in Monkey A. For Monkey D, the particular data shown in Fig. 7f and g did however not include trials with a '594 nm' control but only a 'no laser' control condition. We changed the labels and the colors of the lines accordingly. We also included additional control MUA data for Monkey D from another session in which control stimulation with '594 nm' was directly compared to '505 nm'. The new control data is shown in Supplementary Fig. 4.

Moreover, we improved the statistical analysis that was used to determine which MUA channels are optogenetically modulated (Fig 7b and c). We realized that the statistical test that compares MUA data between baseline and stimulation (-250 to 0 ms vs. 0 to 250 ms) could lead to a false positive result, because the data included some trials in which a visual target stimulus could elicit a small visual response. This can be seen in the yellow trace in Fig. 7c at a time of approximately 70 ms. Therefore, we changed the time window for the statistical test to -25 to 0 ms vs. 0 to 25 ms, relative to the onset of the laser. This window will exclude any potential visual response in area V6 which could lead to a false positive statistical result. However, because the optogenetic effects in the data were very large, the change did not affect the resulting selection of significantly modulated MUA channels.

All changes in the code have been tracked at:

<https://github.com/PJendritza/MultiAreaOptoMarmo>.

Please find below our detailed responses to the reviewer's comments. The color code is as follows:

Blue = reviewer comments

Black = author responses

Orange = text from the revised manuscript

Reviewer #1 (Remarks to the Author):

General comments:

The manuscript by Jendritza and colleagues presents a novel methodology for multi-area recordings and optogenetic stimulation in visual cortex of the marmoset monkey with note worthy results. It presents a methodology that will be of high significance in the emerging marmoset animal model. The manuscript is well written and provides excellent illustrations. The methods are sound and exceed standards in the field. They demonstrate the utility of semi-chronic recordings with silicon linear arrays in marmoset visual cortex both in area V1 and V6. They also introduce methods for optogenetic stimulation and demonstrate in a single monkey that viral expression is was sufficient to modulate performance in a visual detection task. I did not find any major flaws in data analysis or interpretation, although I may suggest a couple additional analyses. Overall, the manuscript provides a valuable resource to the field and note worthy findings.

I have two comments for additional analyses to augment assessment of optogenetic stimulation on behavior: 1) Can the authors more directly determine if the behavioral improvements reflect increases in sensitivity or response bias? 2) Can the authors verify if animals might have learned to detect optogenetic stimulation with the current data, but which is only visible when considering longer response intervals than afforded by the timing control (methods line 656-669) that they implement. Beside those two comments, my other comments are minor and largely for clarity.

Major comments:

Figure 8: Is there an increase in sensitivity in the detection task? While false alarms and hit rates are reported which would suggest d-prime increases, but it is not directly computed. It should be possible to directly computing d-prime for (low con stim) vs (catch trial) as compared to the dprime for (low con + opto) vs (opto only) and confirm if sensitivity increases or if this could be a change in threshold? Related to this, at line 461, authors use the term “behavioral effect”, but it may be possible clarify in the current data if that is an increase in sensitivity or response bias.

We followed the suggestion of the reviewer and performed additional analysis on sensitivity and response bias. The results are included in the main text (lines 444-462) and shown in Supplementary Figure 6.

Supplementary Figure 6 | Behavioral analysis based on signal detection theory. a) Sensitivity (d') and **b)** response bias (c) calculated from 'low contrast visual' vs. 'catch' trials and from 'low contrast + opto' vs. 'opto only' trials. Error bars indicate 95% bootstrap confidence intervals. p-values were calculated via a bootstrap test with 10,000 replications.

We also applied signal detection theory (SDT) to the behavioral data to investigate the effect of optogenetic stimulation on sensitivity and response bias (measured as d' and c , respectively; Stanislaw and Todorov, 1999). We computed d' and c from 'low contrast visual' vs. 'catch' trials and compared them to the values from 'low contrast + opto' vs. 'opto only' trials. Therefore, the 'signal' distributions required for the SDT framework are given by the 'low contrast visual' and by the 'low contrast visual + opto' conditions. Respectively, the 'noise' distributions are given by the 'catch' trials and the 'opto only' trials. Thus, a comparison of d' and c between these conditions allowed us to isolate the effect of optogenetic stimulation during low contrast visual stimulation. Optogenetic stimulation led to an increase in d' (Suppl. Fig. 6a; $p = 0.0367$; bootstrap test), indicating an increase in sensitivity. Optogenetic stimulation also led to a decrease in c from positive to negative values (Suppl. Fig. 6b; $p \leq 0.0001$; bootstrap test), indicating that the animal changed its bias from a 'no' response, i.e. not responding with a saccade ($c > 0$) towards a 'yes' response, i.e. responding with a saccade ($c < 0$). Importantly, while there was no significant difference between the false alarm rate calculated from catch trials and the hit rate from 'opto only' trials (Fig. 8b), it should be noted that the conditions were not rewarded identically. Responses to catch trials were considered false alarms and therefore not rewarded, whereas responses to 'opto only' trials were considered hits and therefore rewarded. Thus, a shift in response bias towards negative c values might be expected because the monkey could increase the number of hits by doing so.

The control for saccade time intervals in computing false alarm rates (for Fig. 8) is interesting, and though reasonable, may deserve brief mention in the main text besides only being mentioned in the methods (lines 656-669), as it is somewhat central to the results presented.

We followed the suggestion of the reviewer and now mention the false alarm recalculation in the results too (lines 395-399):

To prevent false alarms, 40% of all trials were 'catch trials', in which neither an optogenetic nor a visual target appeared. In these trials, the monkey was rewarded for maintaining fixation for 800 ms. In order to avoid overestimation of the false alarm rate due to the longer catch trial duration relative to other trials, we applied a resampling procedure to calculate the true false-alarm rate (see Methods).

In the main text it is reported that "opto only" did not produce more saccades than "catch trials" (marginal effect, $p = 0.0521$); however, this comparison is including a control on timing to match the response intervals to those of go trials with a visual stimulus, correct? This opens a possibility that marmosets did detect the optogenetic stimulation events as compared to catch trials, but made saccades to those opto only events with a slower reaction time than compared to the go trials including a visual stimulus (i.e., the optogenetic event may have been detectable but weaker than a low contrast stimulus). As the authors already note with further training it seems likely marmosets would learn to also detect the optogenetic event (discussion lines 466-468), and possibly if the control for saccade timing is lengthened here, there may already be some evidence of that effect in the current data?

The comparison between "opto only" and "catch trials" does include a control to match the timing between these conditions. However, this timing control only affects catch trials and not any other trial type. Specifically, the timing control does not affect "opto only" trials. This is because only catch trials had a longer duration than the average non-catch trial. Catch trials always lasted 800ms after the onset of the background stimulus. All other trials could last 650-820ms (mean = 735ms). Therefore, the monkey had more time to perform a false alarm during catch trials when compared to other trials. This would lead to an overestimation of the false alarm rate from catch trials. Therefore, we re-analyzed the outcome of the catch trials by matching the catch trial duration with the actual timing of the other conditions, as described in the methods: One randomly selected catch trial with false alarm was compared with the timing of a randomly selected go-trial. If the time of the false alarm from the selected catch trial fell within the time window in which the monkey would have performed a hit (50-500 ms after probe onset), the trial was categorized as a false alarm. If the false alarm timing was such that the monkey would have missed the target, the trial was categorized as correct rejection. Thus, no data is available for non-catch trials that goes beyond what was actually recorded during the experiment because the trial simply ended at the predetermined random interval of 650-820ms. At the end of the trial, the inter-trial interval started with the removal of the fixation point, which could have led to saccades. Therefore, data after trial end cannot be used. This means, we cannot perform the exact analysis suggested by the reviewer.

However, we fully agree with the reviewer that it is interesting to test whether reaction times could differ between opto and non-opto conditions and whether the animal responded with particularly slow reaction times in the "opto only" conditions, at least in the available response window. We therefore performed additional analysis of reaction times. As explained above, we cannot extend

the analysis beyond the end of the trial due to uncontrolled eye movements after the removal of the fixation point. However, the data includes conditions that allow for direct comparison of reaction times. If optogenetic stimulation evoked a detectable but weak signal that would result in a slow reaction time, as suggested by the reviewer, we might expect differences in the RT distributions between “low contrast” and “low contrast + opto” as well as between “opto only” and the sham control. The timing and visual stimulus parameters are identical between these conditions, and the only difference is the presence or absence of effective optogenetic stimulation. Furthermore, the onset timing of the sham stimulation can be used to create a histogram of reaction times when the stimulus was not detectable.

The new analysis shown in Fig. 8c revealed that the reaction times between opto vs. non-opto trials were not significantly different. Furthermore, for the opto only condition, we did not observe a peak at the far side of the reaction time window (<0.5s) that would indicate the monkey was detecting the opto only stimulation. In fact, the reaction time distributions from opto only and sham trials appeared very similar. This indicates that saccades during these conditions were primarily false alarms. However, it should be noted that the statistical power of this analysis is limited due to the number of available trials. Future work should address these issues in detail.

Based on the suggestion of the reviewer, we included the following changes to the manuscript:

Figure 8 | Visual and optogenetic detection task and behavioral results. **a)** Schematic illustration of the detection task. After a brief pre-stimulus fixation period, a background stimulus was shown, followed by the onset of a small visual target with either high or low contrast. On 50% of these trials, the visual target was paired with optogenetic stimulation. Additionally, trials without visual target were included, either with effective laser stimulation (‘Opto only’ condition) or with control laser stimulation, in which the optic fiber was placed outside the craniotomy (‘Sham’ condition), or with no laser stimulation (‘Catch’). All trial conditions except catch trials had identical timing and were rewarded if the monkey executed a saccade 50-500 ms after target or laser onset. **b)** Saccade rates for all task conditions. The animal showed increased detection performance (higher saccade rate) for high-contrast visual targets compared to low contrast targets (93.9% vs. 55.9%, Chi-squared test; $p=3.64e-10$). Pairing high-contrast visual targets with optogenetic stimulation did not result in a difference in saccade rate (Chi-squared test; $p=0.283$). Saccade rate increased significantly when low contrast targets were paired with optogenetic stimulation (55.9% vs. 81.7%; Chi-squared test; $p=1.47e-04$). Optogenetic stimulation alone was not sufficient to be detected by the animal when compared to the false alarm rate (44.5% vs. 33.4%; Chi-squared test; $p=0.0521$). The saccade rate in the sham stimulation control condition (laser fiber positioned 2 mm outside the craniotomy) was not different from the false alarm rate (37.9% vs. 33.4%; Chi-squared test; $p=0.419$).

Number of total trials per condition (hits+misses or correct rejections+false alarms) are shown in parenthesis. Error bars indicate 95% confidence intervals. **c)** Reaction time (RT) distributions from trials with and without optogenetic stimulation for high contrast (top), low contrast (middle) and in the absence of visual stimulation (bottom). No significant differences in RTs were observed (Wilcoxon rank sum test; $p>0.05$). Smooth lines indicate probability density estimates.

Lines 414-418:

Interestingly, saccade rates for optogenetic stimulation alone as compared to the false alarm rate did not quite achieve statistical significance (Chi-squared test; $p=0.0521$; $n=119$ opto only and $n=467$ catch trials). Importantly, the saccade rate in the sham control condition was not different from the false alarm rate (Chi-squared test; $p=0.419$; $n=116$ sham and $n=467$ catch trials).

Lines 426-443:

It is possible that adding optogenetic stimulation results in changes in the animal's reaction time (RT). In the presence of high-contrast visual stimuli, additional optogenetic stimulation might shorten RTs, because the optogenetic stimulation bypasses transmission delays. In the presence of low-contrast visual stimuli, additional optogenetic stimulation might shorten the latency until the combined signal crosses a threshold. In the absence of a visual stimulus, optogenetic stimulation might not lead to significant changes in detection rate (see above comparison of saccade rates in opto-only vs. catch), but it might nevertheless affect RT distributions. To assess these possibilities, we computed RT-histograms and directly compared RTs from trials with and without optogenetic stimulation. Optogenetic stimulation did not result in a significant difference in RTs, neither for high-contrast (Wilcoxon rank sum test; $p=0.4395$; $n=108$ non-opto and $n=109$ opto hit trials), nor for low contrast visual stimulation (Wilcoxon rank sum test; $p=0.1830$; $n=66$ non-opto and $n=89$ opto hit trials). We also compared RTs from "opto only" trials with RTs from sham stimulation. Sham stimulation was timed identical but could not be detected by the animal, thus providing an RT distribution as expected from false alarms with similar number of trials. We did not observe a significant difference in RTs between these conditions (Wilcoxon rank sum test; $p=0.5451$; $n=53$ opto-only and $n=44$ sham hit trials). Furthermore, the RT distribution for 'opto only' trials did not show a peak at the far side of the RT window, which otherwise would indicate that the animal was detecting this type of stimulation at the edge of the RT window.

We also added details on the time windows used in the catch-trial timing control analysis (lines 811-813):

If the time of the false alarm from the selected catch trial fell within the time window in which the monkey would have performed a hit (50-500 ms after probe onset), the trial was categorized as a false alarm.

Minor comments:

In all references: it seems inconsistent if volume number and page numbers are reported for citations.

We fixed the references to be consistent.

Line 76-78: It was unclear to me how the 3D titanium stabilizers minimize the gap between the

bottom of the microdrive and skull? Are the stabilizers screwed into the skull somehow, thus avoiding a gap due to dental cement? This becomes clear, that stabilizers are placed in cement, only later.

We clarified this by adding the following text (lines 78-79) and an additional supplementary figure:

The stabilizers are designed to be positioned very close to the skull, such that they minimize the gap that needs to be filled with cement during implantation (Suppl. Fig. 1). Thus, they make the implantation process easier and faster.

Line 78: "Thus making the implantation process easier and faster" this should be a clause, not a sentence?

This sentence was rewritten in response to the previous question.

Line 236-37: How was a recording site on the array determined to be visually modulated is only mentioned in the Figure 5 caption. It could either be in main text, referenced into methods, or the reference to the figure caption where it is currently described (for the asterisks in 5c) could be more clear in the main text.

We clarified this by adding the following text (line 240-242):

Sites were considered to be visually modulated if MUA between pre-stimulation baseline (-0.25 to 0 s) and post-stimulus time (0 to 0.65 s) was significantly different ($p < 0.05$; Kolmogorov Smirnov test for channels with $MUA > 3\sigma$, see Methods for details).

Line 254: "Note different scaling between b and e" I think that b and g were intended?

We clarified this by adding the following text (line 259-260):

Note different axis scaling and RF sizes between areas V1 and V6 in panel b and e.

Line 285: The smooth fit curves to orientation tuning are Von Mises or Gaussian fits (include a reference to a paper using the curve, or give details in methods).

We clarified this by adding the following text and reference (lines 271-272):

A von Mises function was fit to the mean firing rates for visualization (Berens, 2009).

Line 279: What was the spatial frequency of the stimulus used in Figure 6? As far as I can tell this was never covered in methods (Passive fixation task, line 615) but apparently the spatial frequency was fixed.

We added the spatial frequency information and clarified in which figures the stimulus was used (lines 761-765):

A passive fixation task was used to measure neural responses following visual stimulation with gratings (Fig. 5 and Fig. 6). At the beginning of each trial, the animal was required to maintain its gaze at a central fixation point within a window of 1.4° radius for 100-120 ms. After this period, a static square-wave grating with a spatial frequency of 2 cycles/ $^\circ$ was presented for 650 ms at a Michelson contrast of 80%.

Line 332: Four example units are illustrated in Figure 7 that showed reliable optogenetic activation. What number of neurons in the total population showed similar significant modulations? Was there any specific distribution of which showed stimulation as a function of electrode channel, reflecting any bias for better stimulation at superficial sites possibly? Does light stimulation penetrate to deeper sites?

We followed the advice of the reviewer and performed additional analysis on the dataset from which the examples in Figure 7 were derived. We did not include new datasets. Supplementary Figure 4 shows the population of significantly modulated neurons and their response magnitude as a function of electrode depth. Approximately 50% of the single neurons showed clear optogenetic modulation. Most neurons were recorded in the central part of the electrode array and no obvious depth pattern with regard to stimulation strength was observed.

Supplementary Figure 5 | Single unit population analysis from one optogenetic stimulation session in each animal. a) Scatter plot of single unit activity (SUA) during stimulation with 505 nm laser vs. control condition ($n = 21$ units). Filled symbols indicate significant difference between optogenetic stimulation and control condition ($p < 0.05$, Wilcoxon rank sum test for neurons with $SUA > 3\sigma$ over baseline). Across the two animals, 10 out of 21 units ($\approx 48\%$) showed significant modulation. For the control condition in Monkey A, data from trials with 594 nm stimulation and without laser stimulation were pooled ($n = 180$ control trials; $n = 92$ opto trials). For Monkey D, the control condition included only data without laser stimulation ($n = 84$ control trials; $n = 178$ opto trials). **b)** Magnitude of optogenetic modulation as a function of electrode depth. Most neurons were recorded around the center of the electrode array. No obvious depth pattern with regard to optogenetically induced response strength was observed.

Lines 357-362:

Ten out of the 21 ($\approx 48\%$) well-isolated single neurons from the two example recordings sessions in the two monkeys were significantly modulated ($p < 0.05$, Wilcoxon rank sum test for neurons

with $SUA > 3\sigma$ over baseline, Suppl. Fig. 5a). Most of the recorded neurons clustered around the center of the electrode array. No obvious depth pattern with regard to optogenetically induced response strength was observed (Suppl. Fig. 5b).

Line 367: Was the location of the moving target positioned in the V6 neuron's receptive fields? This is not explicitly stated but I assume it is the case, and it is mentioned in methods but seems relevant to the main text.

We added a sentence for clarification in the main text (lines 388-390):

The center of the movement path was placed within the RF of recording sites with clear optogenetic modulation (see Methods).

And we added several details in the Methods section (lines 782-789).

The center of the movement path of the circular patch was placed at an eccentricity of 8.98° and a polar angle of -72.7° (lower right quadrant). This position fell within the RFs of recording sites on which strong optogenetic modulation was observed. The position was set approximately to the polar angle of the RFs and at the lower end of the area covered by the respective RFs (Suppl. Fig. 7a). This was done in order to make it easier for the animal to perform saccades (head-fixed marmosets have a limited oculomotor range, see Mitchell et al. 2014). RF location and size was determined in a prior recording session from MUA data (see Extended Data Figure 7-1 in Jendritza et al., 2021).

Line 390: Typo: "no" should be "not"

Typo was corrected.

Reviewer #2 (Remarks to the Author):

This manuscript describes novel instrumentation and procedures that allow the simultaneous recording of multiple neurons from two areas of the brain of awake, behaving marmosets, with optional optogenetic stimulation. Recordings are shown to be stable and of high quality. Optogenetic stimulation of area V6 influenced behavior in a near-threshold visual saccade task.

None of the results presented in this paper are particularly surprising, but the technology is impressive and will likely be of broad interest to the readership of Nature Communications. The inclusion of raw data (e.g. Figure 5a and 7a) is a further strength. I am enthusiastic about the work and have only a few minor comments:

Line 77: More information should be provided regarding the 3D printed titanium stabilizers and how they interface with the microdrives. Perhaps a figure (main or supplementary) would help.

We clarified this by adding the following text (lines 78-80) and an additional supplementary figure:

The stabilizers are designed to be positioned very close to the skull, such that they minimize the gap that needs to be filled with cement during implantation (Suppl. Fig. 1). Thus, they make the implantation process easier and faster.

Supplementary Figure 1 | Close-up rendering of the stabilizer and microdrive assembly. The length of the silicon probes ($\approx 8\text{mm}$) results in a gap between the bottom of the microdrive and the skull when the probes are implanted into the superficial parts of the brain. Therefore, a 3D-printed titanium stabilizer was glued to the microdrive prior to the surgery. During the implantation, the assembly can be positioned very close to the skull, such that the gap that needs to be filled with cement is very small.

I was confused regarding surgical implantation of the reference wires. Line 140 indicates that they were implanted during the initial surgery, but line 183 indicates that they were implanted during the second. I think the resolution to this issue is provided in line 540-543, which indicates that two sets of reference wires (backup and actual) were inserted during different surgical procedures. Please clarify.

We clarified this by modifying the main text (lines 142-144):

Two platinum wires were implanted epidurally anterior to the chamber, serving as backup reference wires. The actually used reference wires were implanted subdurally in the second surgery.

Figure 3 legend: “Note the cross-shaped markers on the side of the holder...”. An arrow pointing to the small cross-shaped markers would be helpful.

An arrow was added to Figure 3a.

Are the rasters shown in Figure 6 from a single condition or multiple conditions? Does the variability in the spiking response across trials reflect changes in the visual stimulus or other factors?

The raster plots include data from all orientation conditions. We revised Figure 6 to show trials sorted by stimulus orientation.

An analysis of saccade direction and amplitude would bolster the behavioral component of this study. Did saccade metrics vary by condition? On trials in which optogenetic stimulation was delivered without visual stimulation, and in which the monkey made a saccade within the appropriate time window, were saccades usually directed towards the receptive fields of the stimulated neurons? How about during the catch trials in which the monkey made a false alarm?

We followed the suggestion of the reviewer and performed additional analysis of saccade amplitude and direction. The results of the analysis are included in the main text and in Supplementary Figure 7. The analysis revealed that only visual target contrast had a significant influence on saccade amplitude. Saccade direction did not differ across conditions and was generally consistent with a strategy of performing saccades towards the target location (albeit undershooting in amplitude). This behavior was expected, given that the center location of the moving target was the same across trials.

Supplementary Figure 7 | Saccade analysis of the visual and optogenetic detection task. a) Saccade amplitude for all behavioral conditions. Numbers of saccades per condition are shown in parenthesis. Gray points indicate values from individual saccades. Box plots show medians (red line), 25th and 75th percentile ranges (red filled box) and 1.5 times the interquartile ranges (whiskers). The magenta colored horizontal line indicates the center of the visual target path. Vertical black line on the right shows mean RF extent (data from Jendritza et al. 2021). The animal showed larger saccade amplitude for high-contrast visual targets compared to low contrast targets (Wilcoxon rank sum test; $p=6.66e-10$). Pairing high-contrast or low-contrast visual targets with optogenetic stimulation did not result in a difference in saccade amplitude (Wilcoxon rank sum test; $p=0.427$ and $p=0.219$, respectively). Saccade amplitude from trials with optogenetic stimulation alone and from sham trials did not differ from catch trials (Wilcoxon rank sum test; $p=0.119$ and $p=0.219$, respectively). **b)** Similar to a) but for saccade direction. Saccade direction from trials with high-contrast vs. low contrast visual targets did not significantly differ (Watson's U^2 test; $p=0.0380$). Pairing high-contrast or low-contrast visual targets with optogenetic stimulation did not result in a difference in saccade direction (Watson's U^2 test; $p=0.953$ and $p=0.953$, respectively). Saccade direction from trials with optogenetic stimulation alone and from sham trials did not differ from catch trials (Watson's U^2 test; $p=0.160$ and $p=0.355$, respectively).

Lines 419-425:

We also analyzed the direction and amplitude of the saccades that led to the correct and incorrect responses described above. The analysis revealed that saccade direction did not differ between the tested conditions (Suppl. Fig. 7a). Only the contrast of the visual target but not optogenetic stimulation had a significant effect on saccade amplitude (Suppl. Fig. 7b, Wilcoxon rank sum test; $p=6.66e-10$). Saccades in catch trials were also directed towards the location of the expected target, although no target was shown in these trials. This behavior was expected, given that the center location of the moving target was identical across trials and therefore predictable.

The effect of optogenetic stimulation on saccade probability is reported as not significant, but the p-value, 0.0521, exceeded the arbitrary, conventional threshold of 0.0500 only barely. A phrase like “trending toward significance” or “did not quite achieve statistical significance” might better capture the borderliness of this negative result. This issue reemerges in line 460, which states that “optogenetic stimulation alone was not sufficient to significantly modulate saccade rates”.

We followed the suggestion of the reviewer and changed the text in the Results (lines 414-416):

Interestingly, saccade rates for optogenetic stimulation alone as compared to the false alarm rate did not quite achieve statistical significance (Chi-squared test; $p=0.0521$; $n=119$ opto only and $n=467$ catch trials).

And in the Discussion (lines 528-530):

In contrast, our own results from area V6 indicate that optogenetic stimulation alone was not sufficient to significantly modulate saccade rates, despite being close to the statistical threshold ($p=0.0521$).

Computer aided design materials should be made available.

All 3D designs are available at <https://github.com/PJendritza/Marmo/>. We also added Blender files for the CT segmentation and visualization of the implantation procedure. This is now also reflected in the updated ‘Data and code availability’ statement (lines 880-882):

Code and data to reproduce all figures from this manuscript are available at <https://github.com/PJendritza/MultiAreaOptoMarmo/>. Design files for 3D printing and visualization are available at <https://github.com/PJendritza/Marmo/>.

Reviewer #3 (Remarks to the Author):

This study described a sophisticated and well-described method of preparing marmosets for combined fiber-based optogenetics and multi-electrode electrophysiology. It's a tour de force of methodological engineering in a species that is quite tricky to use. The paper is well written and clear.

Major comment:

The experimental aspects of the study are straightforward and though not themselves groundbreaking, they are excellent and sufficient to establish the success of the method: I have no concerns. It's the hardware development and design, and surgical implantation, that is the critical advance here, especially with respect to the development pathway and implantation of the 3D-printed parts. I felt there was not quite enough detail here before the method to be replicated. I would like to see the authors spell out more fully the entire development cycle pathway for one animal including the step-by-step process of imaging the animal (and segmentation of CT imaging into both hard bony and soft (brain) tissues), the processing and registration of imaging volumes, the rendering choices of blender (and any other software) used for engineering and visualization, including choices made for optimal engineering of the implants as well as the glass brain displays shown in the wonderful figures.

Following the suggestion of the reviewer, we added substantial details to the manuscript. We also restructured the Methods section such that the development cycle from CT imaging, segmentation, implant design and targeting is more clearly laid out. We also added a new section on '3D Rendering' and we made all Blender files that were used for visualization available online for better reproducibility. The main new sections in the manuscript are (lines 593-655):

CT scans, segmentation and alignment

CTs were performed with a Planmeca ProMax 3D Mid scanner (Planmeca Oy, Finland) at 90 kV and 10 mA with a voxel size of 150 μm (isotropic) under brief anesthesia induced with an intramuscular (i.m.) injection of a mixture of alfaxalone (8.75 mg/kg) and diazepam (0.625 mg/kg). The anesthetized animal was placed on a small adjustable bed on which a heating pad was mounted. A plastic head post holder was then used to secure the animal's head in position for the duration of the scan. During CT imaging, animals were not aligned in stereotaxic coordinates. Instead, the chamber and screws were used as fiducial markers for post-scan alignment, as described below. After the scan was completed, CT data was loaded into the 3D Slicer software for segmentation, i.e. delineation of regions in the images that correspond to metal parts, cement and bony tissue. Segmentation was performed by simple intensity thresholding ('Threshold' function in the segment editor of 3D Slicer). Threshold values for upper and lower cutoff were manually set for each animal such that the desired regions of cement, metal or bone were clearly outlined. Specifically, the thresholds for the bony tissue (semi-transparent gray in Fig. 3e-g and Fig. 4e-g) was adjusted until the intracranial space was clearly outlined, such that it could be later used to fit the 3D template brain for coordinate panning. The thresholds for the segmentation of

the chamber and cement were adjusted until the upper rim of the chamber was clearly outlined, such that the segmented region could later be aligned with the 3D model of the chamber. This was important because the chamber served as the reference position for the implantation targets in the brain (see also 3D printed guide in Supplementary Fig. 3). A third threshold-based segmentation was performed that made the four stainless steel screws on the side of the chamber visible. The positions of the four screws were used as additional fiducial markers for alignment to the model of the chamber. All threshold values are listed in Table 2.

Table 2: CT segmentation thresholds:

	Thresholds for chamber/cement	Thresholds for screws	Thresholds for skull
Monkey D:	3206.90 - 30111.00	11326.50 - 30111.00	873.53 - 30111.00
Monkey U:	3206.90 - max	11326.50 - max	873.53 - max
Monkey A:	3206.90 - max	11326.50 - max	1135.69 - max

Segmented volumes were then exported as STL files and imported into Blender for alignment. All CT-based imported volumes were aligned to the 3D model of the chamber. Alignment was performed by manual translation and rotation such that the position of the screws and the upper rim of the chamber in the segmented data aligned with the corresponding positions in the chamber model. Because the chambers had been implanted with the animal aligned in the stereotaxic frame, this effectively brought the CT segmentation data back into stereotaxic coordinates.

Planning of implantation targets

In Monkey A, coordinates for the implantation targets were based on Paxinos et al., 2012. In monkeys D and U, coordinates were based on the following procedure: First, we loaded the MRI-based template marmoset brain segmentation from Liu et al., 2018 into Blender. Specifically, we loaded the segmented volumes of the whole brain (red color in Fig. 3e-g and Fig. 4e-g), and areas V1 and V6 (DM). The MRI data and segmentation can be downloaded at https://marmosetbrainmapping.org/download_atlasv1.html. The template brain (together with area delineations for V1 and V6) was then transformed to fit exactly into the intracranial space of the segmented CT data of each animal. This process was performed manually by translating and scaling in all three spatial dimensions, and rotating in the pitch axis. For better visualization during alignment, we used the ‘clipping border’ function in Blender which allows viewing coronal, sagittal, and horizontal slices of the segmented volumes. Furthermore, for alignment and later visualization, the segmented volume of the skull was set to an alpha value of 0.2 to appear semi-transparent.

This alignment procedure determined where the expected locations of areas V1 and V6 were, relative to the animal's skull and to the already implanted chamber. Thus, for each animal and each area, we selected a target location for implantation based on the individually fitted template brain. For V1, we selected a location close to the midline and close to the border to V2 because this region is known to represent visual space in the lower visual field close to the vertical meridian at an eccentricity of approximately 3-7° of visual angle (Solomon and Rosa, 2014). For V6, the retinotopic map is more complex. Nevertheless, we aimed at targeting the part of V6 with similar, intermediate eccentricities based on the maps presented in Yu et al. 2020. From the resulting

target coordinates for each animal (Table 3), we created a 3D printed implantation guide (stencil) for each animal (Supplementary Fig. 3). In the second surgery, the guide was temporarily placed on the chamber and the two holes pointing at the target locations could be used to mark the positions for the craniotomies in areas V1 and V6.

Table 3: Coordinates of implantation targets:

	V1 caudal from interaural line	V1 lateral from midline	V6 caudal from interaural line	V6 lateral from midline
Monkey D:	7.7 mm	1.3 mm	2.6 mm	4.1 mm
Monkey U:	7.0 mm	1.3 mm	2.2 mm	3.9 mm
Monkey A:	8.5 mm	1.3 mm	2.5 mm	3 mm

3D Rendering

3D renderings were generated via viewport rendering in Blender ('Blender Render' setting) and exported via screen capture with Matlab (code and Blender files are available online, see Data availability). For anti-aliasing, the OpenGL multi sampling setting in Blender was set to a value of 16. For visualization of the implantation sequences (Fig. 3a-d and 4a-d), we used Blender's 'key frames' feature.

The precise choices that must be considered for moving the 3D design to the slicing software and any critical decisions that are required for the metal laser sintering process are key missing details (or at least they are not highlighted well enough for readers to replicate).

As suggested by the reviewer we added details on 3D printing with metal laser sintering technology, including caveats and limitations. The relevant additions and changes can be found in the section '3D printing and implant fabrication' (lines 569-592):

3D printing and implant fabrication

Chambers and microdrive stabilizers were printed via direct metal laser sintering (DMLS) from grade 5 (Ti6Al4V) titanium (Materialise, Belgium). DMLS can produce parts with mechanical and chemical properties comparable to classically CNC machined titanium. However, the minimum feature size is typically 0.4 mm, and the minimum wall thickness is 0.5 mm. This means that very small corners and sharp edges cannot be printed accurately. Thus, the four screw threads (M1.4 thread diameter, 2 mm screw length) that are used to secure the lid to the chamber were manually added after 3D printing. This was done by either using a handheld tapping drill bit or placing the chamber in the CNC mill to create a thread after manual alignment (Chen et al., 2017). Furthermore, due to the sintering process, the surface finish of DMLS parts is rough. Thus, to ensure watertight sealing of the closed chamber, a thin layer of silicone (Kwik-Sil, World Precision Instruments, USA) was applied to the small ridge inside the lid that served as contact area between chamber and lid.

Both, the headpost as well as its holder (Fig. 1d) were produced by standard CNC milling. 3D printing was not viable here, because it does not offer the precision necessary for the fit between headpost and its holder, without substantial post-processing (Chen et al., 2017). All lids were

printed via selective laser sintering from PA12 nylon (Shapeways, USA). Nylon was chosen because of its high abrasion resistance. The use of 3D printed lids makes it possible to rapidly and flexibly produce multiple versions of lids. Before electrode implantation, the inside of the chamber does not contain any parts other than the (optional) reference wires. Therefore, the initial version of the lid was flat and could later be replaced by a taller version. This procedure allowed the animals to gradually get habituated to the size and weight of the final implant. All custom implantation holders and guides were printed from standard resins via stereolithography on a “Form 1” printer (Formlabs Inc., USA).

Besides the adding of threads, which the authors mention is a manual process (which threads and why are they needed, if the implant is fastened to the skull with acrylic cement?), what other post-processing choices are required?

We clarified the process of implant post-processing in the section ‘3D printing and implant fabrication’ that can be found in the previous comment.

A similar sequence of description of the surgical process would also be helpful. For example, it is unclear to me why acrylic cement was chosen over screws to attach the device to the skull. Any caveats to consider with respect to choosing acrylic versus direct hardware attachments to bone?

Details with regard to the use of cement vs. screws to secure the implant to the skull are provided in the section ‘Implantation of chamber and headpost’ in the Methods. For example (line 671-672):

Marmosets have thin skulls and a narrow subdural space, which can make the use of bone screws problematic’.

All critical aspects to consider for the approach, such as cleaning of the skull by mechanical abrasion and H₂O₂ are also described in this section. Nevertheless, regarding the reviewer’s question about caveats of the approach, we added the following sentence (lines 674-676):

To ensure strong attachment of the implant to the skull, we recommend to prepare an area of at least 15x25 mm (Fig. 2b) such that enough surface area is available on which the cement can bond to the skull.

We also added a reference to the section in the relevant part of the main text (lines 140-142):

The skull surface was then cleaned and coated with dental adhesive before a thin layer of cement was applied (Johnston et al., 2018; also see Methods section on ‘Implantation of chamber and headpost’).

REVIEWER COMMENTS

Reviewer #1 (Remarks to the Author):

The authors have addressed my concerns by providing additional clarification and analyses. I believe the manuscript has been significantly improved by the additional analyses and have no further comments to address.

Reviewer #2 (Remarks to the Author):

The authors have done a fine job of revising an already strong manuscript. I have only a few remaining comments.

Figure 7 legend. Missing a right parenthesis and a sentence break. "0 to 25 ms after laser onset...".

Line 422. The text is reversed: Suppl. Fig.7a shows saccade amplitude data and Suppl. Fig.7b shows saccade direction data.

Line 440: typo: "obverse" -> "observe"

Line 442–443: Unclear what "far side of the RT window" means. Please rephrase with temporal language (e.g. early/late) instead of spatial language (e.g. near/far).

Line 453: As I understand it, the d' analysis shows that low contrast + opto stimulation was more detectable than low contrast visual stimulation without the optogenetic stimulation. This does not mean that optogenetic stimulation increased the animal's sensitivity to visual stimulation. Please clarify this, lest the reader misinterpret the result (as I initially did upon reading "indicating and increase in sensitivity").

Lines 838–840: The information provided is insufficient for a reader to implement the bootstrap test. Please provide the additional information that would allow the reader to perform the test.

Reviewer #3 (Remarks to the Author):

The authors did a great job of updating the protocol details following my initial review. The paper was already good, and now I think its fantastic, and will be fully replicable by readers. Its real boon to the field, and I would fully support its publication.

GENERAL REMARK

Please find below our detailed responses to the reviewer's comments. The color code is as follows:

Blue = reviewer comments
Black = author responses
Orange = text from the revised manuscript

REVIEWERS' COMMENTS

Reviewer #1 (Remarks to the Author):

The authors have addressed my concerns by providing additional clarification and analyses. I believe the manuscript has been significantly improved by the additional analyses and have no further comments to address.

Reviewer #2 (Remarks to the Author):

The authors have done a fine job of revising an already strong manuscript. I have only a few remaining comments.

Figure 7 legend. Missing a right parenthesis and a sentence break. "0 to 25 ms after laser onset...".

We fixed the sentence break and added the missing parenthesis.

Line 422. The text is reversed: Suppl. Fig.7a shows saccade amplitude data and Suppl. Fig.7b shows saccade direction data.

We corrected the figure label in the text and changed the order of the sentences accordingly:

The analysis revealed that only the contrast of the visual target but not optogenetic stimulation had a significant effect on saccade amplitude (Suppl. Fig. 7a, Wilcoxon rank sum test; $p=6.66e-10$). Saccade direction did not differ between the tested conditions (Suppl. Fig. 7b).

Line 440: typo: "obverse" -> "observe"

Typo was corrected.

Line 442–443: Unclear what "far side of the RT window" means. Please rephrase with temporal language (e.g. early/late) instead of spatial language (e.g. near/far).

We revised the sentence:

Furthermore, the RT distribution for 'opto only' trials did not show a peak at the late edge of the RT window (0.5 s), which otherwise would indicate that the animal was detecting this type of stimulation but with slow RTs.

Line 453: As I understand it, the d' analysis shows that low contrast + opto stimulation was more detectable than low contrast visual stimulation without the optogenetic stimulation. This does not mean that optogenetic stimulation increased the animal's sensitivity to visual stimulation. Please clarify this, lest the reader misinterpret the result (as I initially did upon reading "indicating and increase in sensitivity").

We agree with the reviewer that an increase in d' between the conditions mentioned above does not necessarily mean that the animal's sensitivity to visual stimulation was changed. It simply means that the sensitivity (as quantified by d') for the detection of paired visual and optogenetic stimulation is higher than the sensitivity of visual stimulation alone. We clarified this in the revised version of the manuscript:

Optogenetic stimulation led to an increase in d' (Suppl. Fig. 6a; $p = 0.0367$; bootstrap test), indicating an increase in sensitivity for the detection of paired visual and optogenetic stimulation when compared to visual stimulation alone.

Lines 838–840: The information provided is insufficient for a reader to implement the bootstrap test. Please provide the additional information that would allow the reader to perform the test.

We added additional information to describe the bootstrap procedures used in the manuscript:

Confidence intervals for d' and c were calculated by a bootstrap procedure across trials (hits, misses, correct rejections and false alarms) with 10,000 bootstrap replications. For each replication, a vector of ones and zeros was generated by sampling with replacement from the original samples of hits and misses (or false alarms and correct rejections, respectively) for each condition. These vectors were then used to calculate d' and c 10,000 times. Thus, the confidence intervals are given by the 5th and 95th percentiles of the resulting distributions. Statistical significance of the differences in d' or c values between different conditions was assessed via calculation of p -values from bootstrapping (Efron and Tibshirani, 1993) with 10,000 replications. For each replication, the difference of the d' or c values between conditions was calculated. The p -value was calculated by finding the proportion of values in the distribution of differences that were equal to or larger than the observed differences between conditions in the sampled data. The smallest possible p -value for 10,000 replications is therefore 0.0001.

Reviewer #3 (Remarks to the Author):

The authors did a great job of updating the protocol details following my initial review. The paper was already good, and now I think its fantastic, and will be fully replicable by readers. Its real boon to the field, and I would fully support its publication.